# Clinical interpretation of variants identified in *RNU4ATAC*, a non-coding spliceosomal gene

Clara Benoit-Pilven[1,2☯¤], Alicia Besson[1☯], Audrey Putoux[1,3], Claire Benetollo[4], Clément Saccaro[1], Justine Guguin[1], Gabriel Sala[1], Audric Cologne[1,2], Marion Delous[1], Gaetan Lesca[1,3], Richard A. Padgett[5], Anne-Louise Leutenegger[6], Vincent Lacroix[2], Patrick Edery[1,3‡], Sylvie Mazoyer[1‡]*

1 Equipe GENDEV, Centre de Recherche en Neurosciences de Lyon, Inserm U1028, CNRS UMR5292, Université Lyon 1, Université St Etienne, Lyon, France, 2 Laboratoire de Biométrie et Biologie Évolutive, CNRS UMR5558, Université Lyon 1, Villeurbanne, and EPI ERABLE - Inria Grenoble, Villeurbanne, Rhône-Alpes, France, 3 Service de génétique et Centre de Référence Maladies Rares des Anomalies du Développement CLAD Sud-Est, Hospices Civils de Lyon, Lyon, France, 4 Plateau NeuroGénétique FOnctionnelle, Centre de Recherche en Neurosciences de Lyon, Inserm U1028, CNRS UMR5292, Université Lyon 1, Université St Etienne, Lyon, France, 5 Department of Cardiovascular and Metabolic Sciences, Lerner Research Institute, Cleveland Clinic, Cleveland, OH, United States of America, 6 Université de Paris, NeuroDiderot, Inserm, Paris, France

☯ These authors contributed equally to this work.
¤ Current address: Institute for Molecular Medicine Finland (FIMM), University of Helsinki, Helsinki, Finland
‡ These authors also contributed equally to this work.
* sylvie.mazoyer@inserm.fr

**Data Availability Statement:** All relevant data are within the manuscript and its Supporting Information files.

## Abstract

Biallelic variants in *RNU4ATAC*, a non-coding gene transcribed into the minor spliceosome component U4atac snRNA, are responsible for three rare recessive developmental diseases, namely Taybi-Linder/MOPD1, Roifman and Lowry-Wood syndromes. Next-generation sequencing of clinically heterogeneous cohorts (children with either a suspected genetic disorder or a congenital microcephaly) recently identified mutations in this gene, illustrating how profoundly these technologies are modifying genetic testing and assessment. As *RNU4ATAC* has a single non-coding exon, the bioinformatic prediction algorithms assessing the effect of sequence variants on splicing or protein function are irrelevant, which makes variant interpretation challenging to molecular diagnostic laboratories. In order to facilitate and improve clinical diagnostic assessment and genetic counseling, we present i) an update of the previously reported *RNU4ATAC* mutations and an analysis of the genetic variations affecting this gene using the Genome Aggregation Database (gnomAD) resource; ii) the pathogenicity prediction performances of scores computed based on an RNA structure prediction tool and of those produced by the Combined Annotation Dependent Depletion tool for the 285 *RNU4ATAC* variants identified in patients or in large-scale sequencing projects; iii) a method, based on a cellular assay, that allows to measure the effect of *RNU4ATAC* variants on splicing efficiency of a minor (U12-type) reporter intron. Lastly, the concordance of bioinformatic predictions and cellular assay results was investigated.

**Funding:** This work was supported by CNRS, Inserm, Université Paris 7 and Université Lyon 1 through recurrent funding, the ANR Aster (no. ANR-16-CE23-0001) and U4ATAC-BRAIN (no. ANR-18CE12-0007-01) grants and an Inserm/ Hospices Civils de Lyon grant to P.E. (Contrat d'Interface pour Hospitaliers). A.C. was supported by a grant from Inria (Thèse Inria-Inserm "Médecine Numérique" - 2016) and C.B.P. by a grant from the Fondation pour la Recherche Médicale to P.E. (Financement d'un ingénieur - ING20160435660). The funders had no role in study design, data collection and analysis, decision to publish, or preparation of the manuscript.

**Competing interests:** The authors have declared that no competing interests exist.

## Introduction

The sequences of thousands of genes involved in the aetiology of one or several Mendelian genetic diseases are routinely evaluated in patients in order to provide or confirm their diagnosis and help managing their health care. DNA variant interpretation is currently one of the major challenges of genetic testing. Indeed, diagnostic laboratories have seen a massive increase in the number of variants identified due to the widespread implementation of next-generation sequencing. Recommendations for the homogenised setting of variant classification pipelines have been published: they take into account variant characteristics (i.e. the type of variant: missense, nonsense, indel, splice site; sequence conservation among species; *in silico* predicted consequence), epidemiological and segregation data, and functional evaluation [1]. Most genes involved in Mendelian diseases are protein-coding genes and therefore the main features of these recommendations apply to them, despite their many different biological properties. However, there exist a few rare diseases for which pathogenic variants have been identified in a handful of non-coding genes, one of them being the snRNA gene *RNU4ATAC* [2].

*RNU4ATAC* was first found mutated in an autosomal recessive disorder named microcephalic osteodysplastic primordial dwarfism type 1 (MOPD1, OMIM 210710) or Taybi-Linder syndrome (TALS) [3,4]. This very rare (~50 reported cases worldwide) and severe disorder is characterized by intellectual disability and multiple malformations including severe microcephaly, cortical brain malformations (neuronal migration defects), corpus callosum agenesis/ dysgenesis, dysmorphic features, dwarfism, and bone anomalies. It leads to early unexplained death occurring within the first two years of life in more than 70% of the published cases. Other very rare congenital disorders named Roifman syndrome (RFMN, OMIM 616651) [5] and Lowry Wood syndrome (LWS, OMIM 226960) [6] have subsequently been attributed to biallelic *RNU4ATAC* mutations. Both RFMN and LWS have features overlapping with TALS (i.e. microcephaly, growth retardation, skeletal dysplasia, intellectual disability). However, severe structural brain anomalies and early death are not observed in these two latter disorders, and microcephaly and growth retardation are less pronounced [7]. On the other hand, because RFMN patients' parents first consult because of their child's recurrent infections, immune defects have been thoroughly investigated and are well documented in this syndrome, whereas this is not the case for TALS and LWS.

Most small nuclear RNAs (snRNAs) are components of either the major and/or the minor spliceosome, which respectively removes major (also called U2-type) or minor (U12-type) introns from pre-mRNAs. Pre-mRNA splicing is a crucial step in the expression of eukaryotic genes, especially in humans where about 97% of the ~20.000 protein-coding genes contain at least one intron. Major introns represent more than 99% of the total number of introns (~220.000), while there are only ~850 minor introns present in about 700 genes [8–10]. The two spliceosomes are highly homologous and owe their specificity to different consensus splice-site sequences as well as different sets of snRNAs, i.e. U1, U2, U4 and U6 for the major spliceosome and U11, U12, U4atac and U6atac for the minor one [11]. On the other hand, U5 snRNA and the protein components are common to both spliceosomes, apart from seven proteins specific to the minor spliceosome. Homologous snRNAs with equivalent functions in the two spliceosomes, namely U1/U2 and U11/U12, U4/U6 and U4atac/U6atac, share a common secondary structure formed by intra- and/or intermolecular base-pairing despite their divergent nucleic acid sequences. The U4atac/U6atac small nuclear ribonucleoprotein particle (U4atac/U6atac di-snRNP) is composed of U4atac snRNA stably base-paired with U6atac snRNA and of seven Sm proteins and other particle-specific proteins. This di-snRNP then associates with the U5 snRNP, forming the U4atac-U6atac.U5 tri-snRNP, a component of the pre-catalytic complex which gains its catalytic activity to excise the intron by the withdrawal of

U4atac and the pairing of U6atac with U12 [11]. Several *RNU4ATAC* mutations identified in TALS patients have been shown to result in defects in minor tri-snRNP formation [12]. Further, transcriptomic analyses of cells from RFMN and TALS patients revealed massive U12 intron retentions [5,9,13,14].

Even though TALS, RFMN and LWS syndromes are extremely rare, biallelic *RNU4ATAC* mutations have been recently identified during the screening of clinically heterogeneous cohorts. Indeed, one carrier was found in a whole genome sequencing analysis of 103 patients from pediatric non-genetic subspecialty clinics, each with a clinical phenotype suggestive of an underlying genetic disorder, yet undiagnosed [15]. Another one was found, less unexpectedly, in a gene panel or exome sequencing of 150 patients (104 families) with Mendelian forms of congenital microcephaly (occipital frontal circumference below -2 SD or with a reported history of microcephaly at birth) [16]. Genotypes and phenotypes of these two cases were compatible with RFMN and TALS syndromes, respectively. The generalisation of the use of exome and whole-genome sequencing to diagnose disorders with a suspected genetic cause implies that new *RNU4ATAC* variants will undoubtedly be identified in the future in laboratories without expertise on this gene. Yet, the classification of *RNU4ATAC* variants to provide accurate genetic counselling is difficult because the criteria used to predict the impact of variants in coding genes are partially inappropriate. Furthermore, functional assays are lacking and no published guidelines are yet available. In order to facilitate variant interpretation for diagnostic laboratories, we first reviewed the literature and compiled the *RNU4ATAC* variants reported as pathogenic in patients at the homozygous or compound heterozygous state. We next analysed the extent of genetic variations found in this gene using the Genome Aggregation Database (gnomAD) resource. We further predicted the impact of all of these variants on the secondary structure of the U4atac/U6atac bimolecule using an available bioinformatic tool, RNAstructure, and with the widely used Combined Annotation Dependent Depletion (CADD) tool. In addition, we assessed the splicing efficiency of U4atac molecules carrying either one of 24 variants using a cellular model. We then compared the scores obtained with RNAstructure and CADD with the results obtained with the cellular assay and confronted them to their known pathogenic status.

## Methods

### Variant information extraction from gnomAD

*RNU4ATAC* variant information was extracted from the Genome Aggregation Database v2.1 (gnomAD, http://gnomad.broadinstitute.org/) [17]. We kept the variants whose coordinates were comprised in the following interval: chr2:122,288,456–122,288,585 (GrCh37). Due to the misannotation of the *RNU4ATAC* sequence in gnomAD, we had to correct the names of the variants by adding 1 to the coordinate of the *RNU4ATAC* nucleotide (e.g. n.50G>A was transformed into n.51G>A).

### Bioinformatics model for large-scale U4atac/U6atac secondary structure predictions

Three nucleotide substitutions found in patients but not present in gnomAD were added to the list, leading to a total number of 285 different *RNU4ATAC* variants for the 130 nucleotides of the *RNU4ATAC* gene. The bioinformatics pipeline that we set-up follows three steps. First, we generated the mutated *RNU4ATAC* sequences for the 285 variants. Then, for all the obtained sequences, we used the *bifold* function of the RNAstructure package (version 6.1) [18] to predict the secondary structure of the bimolecule formed by wild type U6atac and

mutant U4atac. This tool predicts the bimolecular structures of two sequences folded into their lowest hybrid free energy conformation. All default parameters were used to run *bifold* except the "-p" parameter that we set to 2. This option allowed us to keep all suboptimal structures with a free energy close to the energy of the minimal free energy (MFE) structure. Finally, the predicted bimolecules were compared to the wild type MFE structure to compute a score based on the number of base-pairing changes due to the presence of the variants and the known importance for splicing of the regions affected by the change of structure: score = A x 3 + B x 1 + C x 0.5, where A, B and C are the numbers of modified base pairings in regions of major, variable or limited/null importance for splicing respectively, as defined by Merico et al. [5]. This score was computed for all suboptimal structures corresponding to each mutant U4atac, allowing us to choose among the suboptimal structures the closest one to the structure of wild type U4atac/wild type U6atac. The final output is the list of *RNU4ATAC* variants with their associated score. The scripts to run this analysis are available on github: https://github.com/cbenoitp/RNU4atac_variants.

## Combined Annotation Dependent Depletion (CADD) predictions

The CADD score combines the results of >60 variant annotation prediction tools into one metric by contrasting variants that survived natural selection with simulated mutations, hence representing a measure of deleteriousness for single nucleotide variants and small indels (version v1.5 at https://cadd.gs.washington.edu/). A low score indicates that a variant resembles commonly occurring genetic variation that poses no apparent disadvantage for an organism. In contrast, a high score represents variants that are more likely to have deleterious effects [19]. In this study, we used "raw" C-scores rather than the "scaled" ones as the comparison with the scores obtained with variants in protein-coding genes appeared of little value.

## Cellular model

**Plasmids.** The P120 minigene reporter plasmid was constructed by R. Padgett and coll. [20]. Briefly, it derives from the pCB6 expression vector into which a portion of the human *NOP2* (*NOP2 Nucleolar protein)* gene has been inserted downstream the CMV promoter. This portion consists of parts of exons 5 and 8, and all of exons 6 and 7 and introns E, F (a U12-type intron) and G. In the U4atac expressing vector also constructed by R. Padgett and coll., the human U4atac snRNA sequence replaces the U1 snRNA sequence of a functional U1 snRNA gene cloned into the pUC13 expression vector [21].

**Cells.** Primary fibroblasts were derived from skin biopsies of a control child or from a TALS patient carrying the *RNU4ATAC* n.51G>A pathogenic variant in the homozygous state (TALS2 in [3]). Informed written consent for the use of these samples in research was obtained from parents of the TALS patient and control child. Cells were collected, processed, and stored in Lyon University Hospital biobank (CBC Biotec). Authorisation for their collection and their use in research has been granted by the Ministry of Research, by the Comité de protection des Personnes Sud-Est IV and the Regional Agency for Hospital Services under the number DC-2015-2566. The project has been approved by the local ethics committee of the Hospices Civils de Lyon.

**Transfection.** Transient transfection of the P120 minigene and U4atac snRNA expression plasmids into cultured n.51G>A TALS patient fibroblast cells was performed using Lipofectamine® 2000 Reagent (Thermo Fisher Scientific) according to the manufacturer's protocol. For these experiments, 0.06 μg of P120, 1.25 μg of the U4atac snRNA expression plasmid, and 1.19 μg of empty pUC19 plasmid (added to keep a total plasmid DNA quantity of 2.5 μg as required for an optimal transfection) were added to 250,000 cells /well of a 6-well plate. For

testing the joint effect of two different variants, co-transfections of 0.625 μg of each version of U4atac snRNA expression plasmids was realised. Every transfection experiment performed to test a batch of *RNU4ATAC* variants included a set of cells transfected with the WT U4atac snRNA expression plasmid.

**RNA extraction.** Total RNA was isolated from cells 48h post-transfection using the Nucleospin RNA kit (Macherey Nagel) according to the manufacturer's instructions. A RNA Qualified-DNase treatment (RQ1 DNase, Promega) was performed to remove contaminating genomic DNA from RNA samples according to manufacturer's protocol.

**Reverse transcription.** 200 ng of RNA was reverse transcribed using the GoScript Reverse Transcriptase kit (Promega) and a reverse primer specific of P120 minigene construct (R: 5′- GGA TCC TCT AGA GTC GAC C-3′), which allows to target the transcripts produced from the P120 plasmid specifically, without amplifying the endogenous *NOP2* transcripts.

**Semi-quantitative RT-PCR.** Semi-quantitative RT-PCR were performed using the Platinum Taq DNA polymerase (Thermo Fisher Scientific) according to the manufacturer's instructions and a primer pair flanking the P120 U12 intron (intron F) (F: 5′-TGA GGA ACC ATT TGT GCT GC-3′, R: 5′-ATC CGC TTG TGA ACT CGT TG-3′). The PCR products were separated on a 2% agarose gel electrophoresis using GelRed nucleic acid gel stain (Biotium). The PCR product intensity was quantified using Image J. Splicing efficiency was measured as the ratio of spliced RNA to spliced RNA and unspliced RNA (expressed as a percentage). In addition, RT-PCR were performed with cDNA made from RNA of untransfected cells to check the absence of endogenous *NOP2* gene amplification. A negative control (RT-PCR without reverse transcriptase) was also performed to check the absence of genomic DNA in RNA samples.

**Quantitative RT-PCR.** qRT-PCR were performed in triplicate using Rotor-Gene SYBR Green PCR kit (Qiagen) according to the manufacturer's instructions. For these experiments, three primer pairs were used to measure relative splicing efficiency: i) primer pair which targets P120 transcripts with unspliced intron F (F: 5′-TGA GGA ACC ATT TGT GCT GC- 3′, R: 5′-GGA AAT CCC TCT CCC AAC C-3′); ii) primer pair which targets P120 transcripts with spliced intron F (F: 5′-GGA GAT GGA GCA GGA TGC-3′, R: 5′-TCCCGCT GAGCCCCAAAA-3′); iii) primer pair which targets all P120 transcripts (F: 5′-CAG ACC TGC AAC GAG TTC AC-3′, R: 5′-GTA TTC AGA ACG AGA CCG CC-3′). The 2–ΔΔCt method was used to measure relative splicing efficiency by qPCR. We computed the fold change of number of P120 unspliced and spliced transcripts in mutant U4atac snRNA condition relative to wild-type U4atac snRNA condition, normalized to the number of P120 total transcripts. Then, the relative splicing efficiency was calculated as the ratio of spliced RNA to unspliced and spliced RNA. The relative splicing efficiency in wild-type U4atac snRNA condition was fixed at 100%. Relative quantification of U4atac snRNA expression were performed using qRT-PCR and the primer pair (F: 5′-AAC CAT CCT TTT CTT GGG GTT GC-3′, R: 5′-ATT TTT CCA AAA ATT GCA CCA AAA TAA AGC-3′). For each variant, the significance of the difference between the relative splicing efficiency and 100% was tested using a one-tailed t test.

## Results

### Inventory of the disease-associated *RNU4ATAC* variants

To date, biallelic *RNU4ATAC* pathogenic variants have been reported in the literature in 46 families: 31 TALS (46 patients and 8 foetuses), 11 RFMN (15 patients) and 4 LWS families (5 patients) (S1 Table). Half of these independent occurrences (23/46) are due to homozygous *RNU4ATAC* variants which are (or are suspected to be) the result of consanguineous unions.

Among the 30 different pathogenic variants identified, 29 are substitutions concerning 23 of the 131 U4atac nucleotides, and the remaining one is an 85-nt duplication (Table 1). As shown in Fig 1, the single nucleotide variants found in U4atac are mostly located in three regions of the U4atac/U6atac bimolecule (26/30): the intramolecular 5' Stem-Loop structure (12 substitutions), the region forming the Stem II structure through interaction with U6atac (7 substitutions), and the single-stranded Sm protein-binding region (7 substitutions). Unsurprisingly, these regions have been characterised as the most important ones for U4atac maturation and function [22,23]. The 5' Stem-Loop exhibits a structural K-turn motif that interacts with the 15.5K protein, then inducing the association of PRPF31 protein to the U4atac/U6atac-15.5K complex and, in turn, the binding of the PRPF31/PRPF4/PPIH heterodimer to Stem II [22–25]. The resulting snRNP complex is essential for the activation of the spliceosome prior to catalysis. The Sm protein-binding region present in every snRNA but U6 and U6atac, allows the interaction with seven Sm proteins required for the snRNA maturation [26]. Three variants are located outside of these three functionally important regions, namely a substitution that resides just 3' to Stem I and two substitutions located in the intramolecular 3' Stem-Loop. The 3' Stem-Loop is not known to interact with any protein and has been shown to be less important for splicing than the 5' Stem-Loop using cellular assays, but it is nevertheless suspected to function as part of the Sm protein-binding signal [22].

The most frequent pathogenic variant, n.51G>A, was identified in 22 out of the 46 *RNU4A-TAC* families, particularly in TALS families (20/31) whether they reside in Africa, Europe, Middle East, North America, or Asia. Fourteen of them, presenting the most severe form of TALS (death occurring before 2.5 years of age), carry n.51G>A in the homozygous state, while the remaining six TALS families carry various variants on the other allele. The n.51G>A variant was also found in the compound heterozygous state in one RFMN and one LWS family. Only one other variant was found in all three *RNU4ATAC*-associated syndromes, namely n.46G>A (one TALS, one RFMN and one LWS families). Three additional variants, n.29T>C, n.111G>A, and n.116A>T were found in two syndromes (TALS and RFMN for the former and the latter, TALS and LWS for the other one), while the other variants are restricted to one phenotype so far (S1 Table).

Attempts at correlating *RNU4ATAC* genotype with disease phenotype are hampered by the small number of patients, especially in the cases of RFMN and LWS. It is nevertheless striking to notice that 25 out of the 31 TALS families carry homozygous or compound heterozygous mutations in the 5' Stem-Loop (20 and 5 families respectively), while another 5 families with compound heterozygous mutations carry one mutation in this region and the other one in either the Sm protein-binding site (3 families) or the 3' Stem-Loop (1 family); in one family, the second mutation is a large duplication. The only TALS family without mutation in the 5' Stem-Loop carries a mutation just one nucleotide 3' to Stem I and the other in the 3' Stem-Loop (S1 Table). Early lethality appears to be associated with n.51G>A, although not systematically (one n.51G>A;n.55G>A child was still alive at 6 years-old), and to lesser extents to n.55G>A, n.50G>C and n.50G>A. Concerning RFMN families, it is also striking that all 11 of them carry one mutation in Stem II, a region which has never been found mutated in TALS patients, as noted in the princeps paper [5]. Two among them are homozygous for a mutation in Stem II, n.16G>A, while the others carry either a second mutation in the 5' Stem-Loop (5 families) or in the Sm protein-binding site (4 families) (S1 Table). No clear pattern is seen for the 4 LWS families, which present with diverse combinations of *RNU4ATAC* mutation location (S1 Table).

## *RNU4ATAC* variants identified in large-scale sequencing projects

To gain insight into the extent of *RNU4ATAC* genetic variability, we took advantage of the Genome Aggregation Database (gnomAD) resource (http://gnomad.broadinstitute.org/),

**Table 1. Summary of the published disease-associated *RNU4ATAC* variants.**

| Variant | RNA domain | Number of independent patient(s) (homozygous + compound heterozygous states) | Origin of the patients (number of independent cases) | References |
|---|---|---|---|---|
| n.5A>C | Stem II | 1 LWS (0+1) | ? (1) | [6] |
| n.8C>A | Stem II | 1 LWS (0+1) | Italian ? (1) | [7] |
| n.8C>T | Stem II | 1 RFMN (0+1) | Albanian (1) | [5] |
| n.13C>T | Stem II | 4 RFMN (0+4) | English (1), Italian (1), Belgian (1), ? (1) | [5,37,38] |
| n.13C>G | Stem II | 1 RFMN (0+1) | ? (1) | [15] |
| n.16_100dup | | 1 TALS (0+1) | Danish (1) | [34] |
| n.16G>A | Stem II | 6 RFMN (4+2) | Lebanese (1), Pakistani (1), Belgian (2) | [5,13,14] |
| n.17G>A | Stem II | 1 RFMN (0+1) | Tamil (1) | [13] |
| n.29T>C | 5' Stem-Loop critical region | 1 TALS (0+1) | Chinese (1) | [15] |
| | | 1 RFMN (0+1) | ? (1) | [39] |
| n.30G>A | 5' Stem loop critical region | 1 TALS (0+1) | German (1) | [4] |
| n.37G>A | 5' Stem loop critical region | 1 RFMN (0+1) | English (1), Italian (1) | [5] |
| n.40C>T | NOP domain binding region | 2 TALS (0+2) | Danish (1), French (1) | [34,40] |
| n.46G>A | | 1 TALS (1+0) | Turkish (1) | [41] |
| | | 1 LWS (0+1—in cis with n.123G>A) | ? (1) | [6] |
| | | 1 RFMN (0+1) | Belgian (1) | [14] |
| n.48G>A | 5' Stem loop critical region | 1 RFMN (0+1) | Italian (1) | [5] |
| n.50G>C | 5' Stem loop critical region | 1 TALS (0+1) | North American (1) | [3] |
| n.50G>A | 5' Stem loop critical region | 1 TALS (0+1) | North American (1) | [3] |
| n.51G>A | 5' Stem loop critical region | 20 TALS (14+6) | Algerian (1), Turkish (2), Moroccan (2), Indian (2), North American (4), Norwegian (1), Maltese (2), Rwandan (1), French (1), Deutch (1), Chinese (1), Egyptian (2) | [3,4,39,40,42,43] |
| | | 1 RFMN (0+1) | Lebanese (1) | [5] |
| | | 1 LWS (0+1) | ? (1) | [6] |
| n.53C>G | 5' Stem loop critical region | 1 TALS (0+1) | Norwegian (1) | [3] |
| n.53C>T | 5' Stem loop critical region | 1 LWS (0+1) | Italian ? (1) | [7] |
| n.55G>A | 5' Stem loop critical region | 6 TALS (4+2) | German (1), Egyptian (2), Yemeni (1), Indian (1), ? (1) | [4,16,35,44, 45] |
| n.66G>C | | 1 TALS (0+1) | Egyptian (1) | [45] |
| n.111G>A | 3' Stem loop | 1 TALS (0+1) | German (1) | [4] |
| | | 1 LWS (0+1) | ? (1) | [6] |
| n.114G>C | 3' Stem loop | 1 LWS (0+1) | Italian ? (1) | [7] |
| n.116A>T | Sm protein-binding site | 1 TALS (0+1) | Belgian (1), ?(1) | [16,37] |
| | | 1 RFMN (0+1) | | |
| n.116A>G | Sm protein-binding site | 1 RFMN (0+1) | Tamil (1) | [13] |
| n.116A>C | Sm protein-binding site | 1 RFMN (0+1) | German (1) | [38] |
| n.118T>C | Sm protein-binding site | 1 RFMN (0+1) | Albanian (1) | [5] |

(*Continued*)

**Table 1.** (Continued)

| Variant | RNA domain | Number of independent patient(s) (homozygous + compound heterozygous states) | Origin of the patients (number of independent cases) | References |
|---------|-----------|----------------------------------------------|----------------------------------------|------------|
| n.120T>G | Sm protein-binding site | 1 LWS (0+1) | Italian ? (1) | [7] |
| n.123G>A | Sm protein-binding site | 1 LWS (0+1 –in cis with n.46G>A) | ? (1) | [6] |
| n.124G>A | Sm protein-binding site | 3 TALS (0+3) | Egyptian (1), French (1), Deutch (1) | [40, 45] |

LWS: Lowry-Wood syndrome; RFMN: Roifman syndrome; TALS: Taybi-Linder syndrome

exploiting the data present in the v2.1.1 version derived from 15,708 whole-genome and 65,258 exome sequences [17,27]. Of note, *RNU4ATAC* is half as much covered by exome sequencing as most coding genes because some commercially available exome capture kits target only a fraction of all non-coding genes.

We listed 282 variants in *RNU4ATAC*: 239 substitutions concerning 123 of its 130 nucleotide-long genomic sequence, as well as 19 insertions, 16 duplications and 8 deletions whose size range from 1 to 107 (S2 Table). Nearly half of these variants (129/282; 46%) were found in only one or two of the ~81.000 screened individuals [allele count: 1–465, median = 3]. The most frequent variant identified, n.23C>T (never identified in patients), is present in only 0.29% of the screened alleles, suggesting a strong selective pressure against variations in this gene. This is consistent with the fact that genes encoding components of complexes such as the spliceosome, ribosome and proteasome involved in core biological processes are the most constrained ones [27]. However, when looking into sub-population allelic frequencies, 23 variants have a frequency > 0.1% in at least one sub-population (S2 Table), and three have a

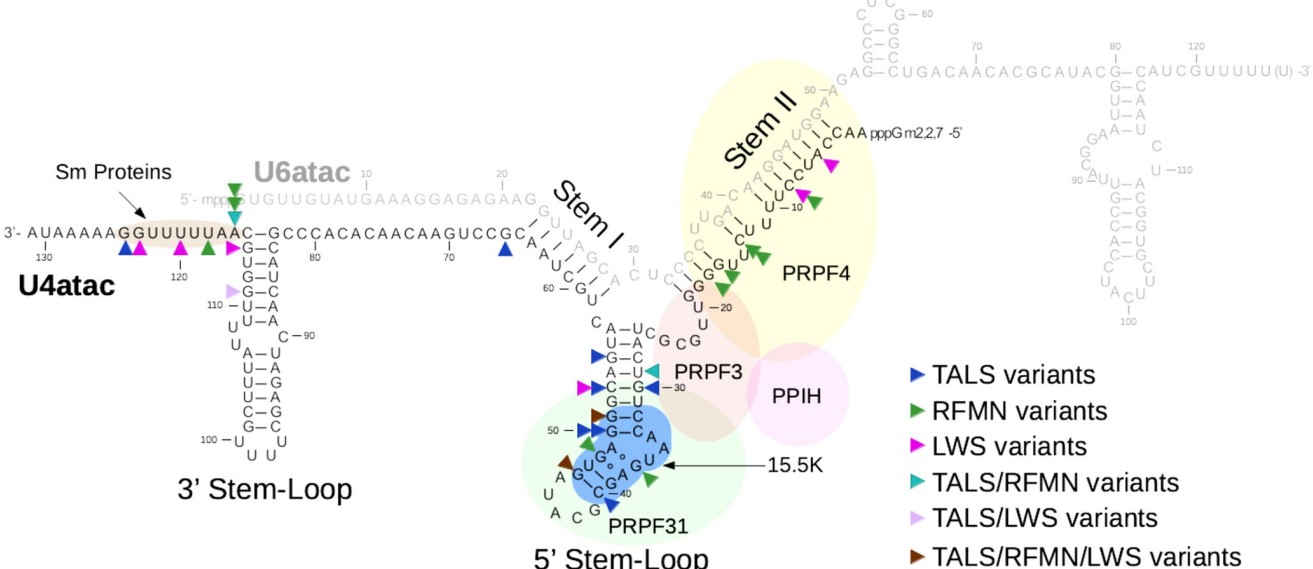

**Fig 1. U4atac nucleotides mutated in TALS, RFMN and LWS patients.** The U4atac/U6atac bimolecule is represented, as well as its main interacting proteins (15.5K, PRPF3, PRPF4, PRPF31, PPIH, and the Sm proteins). Arrowheads point to the mutated nucleotides observed in TALS, RFMN and/or LWS. There are five nucleotides for which more than one substitution was identified (more than one arrowhead in the figure). The 85 nt-duplication (n.16_100dup) is not shown.

frequency > 1% and are found in the homozygous state, namely n.58C>T (1.88% in Africans; two homozygous individuals), n.87C>T (2.99% in Ashkenazi Jewishs; one homozygous individual), and n.93G>A (1.19% in South Asians; six homozygous individuals) (S2 Table). Despite lower allele frequencies, three more variants were found in the homozygous state: n.110delT (0.23% in South Asians; two homozygotes), n.91dupT (0.29% in Latinos; one homozygote) and n.45A>G (0.01% in South Asians; one homozygote) (S2 Table). Among the six variants found in the homozygous state, the most unlikely to be pathogenic, four are located in the 3' Stem-Loop in a region considered of either limited or null importance for splicing (n.93G>A) or variable importance (n.87C>T, n.91dupT and n.110delT); n.58C>T lies between the 5' Stem-Loop and Stem I, and n.45A>G is in the loop of the 5' Stem-Loop, both regions also considered to be of limited or null importance for splicing [22].

Among the 30 variants reported as pathogenic, 27 were identified in large-scale sequencing projects and have allele frequency ranging from 0.0008% to 0.04% for the most frequent pathogenic variant, i.e. n.51G>A. Noteworthily, n.51G>A is the 20th most frequent *RNU4ATAC* variant among all those present in gnomAD and it has been identified in all sub-populations but one (Ashkenazi Jewish) at frequencies ranging from 0.03% to 0.06% (S2 Table).

In summary, population data showed that although the extent of variations in *RNU4ATAC* is extremely large, all of them classify as rare variants.

## Bioinformatics predictions

*RNU4ATAC* is a single exon non-coding gene and the functional consequences of its variants cannot be predicted using traditional tools which focus on amino acid substitutions or splicing variants. It is however possible to predict the impact of the variants on the secondary structure of the U4atac/U6atac bimolecule using a dedicated software, RNAstructure [18], the only tool predicting the correct structure, to date. Using the bifold function of RNAstructure (http://rna.urmc.rochester.edu/RNAstructure.html) in an automated way which we set up in house (Fig 2A), we modelled the effect of the 282 variants identified in large-scale sequencing projects as well as the three pathogenic variants not present in gnomAD, namely n.8C>T, n.120T>G and n.16_100dup. To assess the structural modifications of the U4atac/U6atac bimolecule resulting from U4atac variants, we defined a scoring system that takes into account the number of base-pairing changes and the importance for splicing of the U4atac snRNA domain where they occur (Fig 2B). The distribution of the produced scores is shown in Fig 3A. Among the 30 disease-associated *RNU4ATAC* variants (S3 Table), 13 had a score ≥ 10 (43%) and 20 ≥ 1 (67%); among the 255 other variants, 50 had a score ≥ 10 (20%) and 138 ≥ 1 (54%). The 10 disease-associated variants with a null score are located in the Sm protein-binding region (6 variants), in Stem II (2 variants), and in the non-canonical base pair forming across the pentaloop of the 5' Stem-Loop which allows U4atac to establish more intimate interactions with PRPF31 (2 variants). Among the 6 variants found in the homozygous state in large-scale sequencing projects, only one modifies the structure of the bimolecule according to RNAstructure, n.110delT, which has a score of 5 and resides within the central loop of the 3' Stem-Loop, a region shown in a cellular assay to be of variable importance for splicing [22]. In summary, these findings show that using only predictions of the modification of the structure of the bimolecule made with RNAstructure suffer from a lack of sensitivity for predicting *RNU4ATAC* variant pathogenicity, as 10 of the 30 variants identified in patients are not found to have structural consequences.

We also tested the Combined Annotation Dependent Depletion (CADD) tool equipped to handle both coding and non-coding variants, contrary to most performing tools that are protein-based and focus on non-synonymous variants [19,28]. The CADD scores ranged from

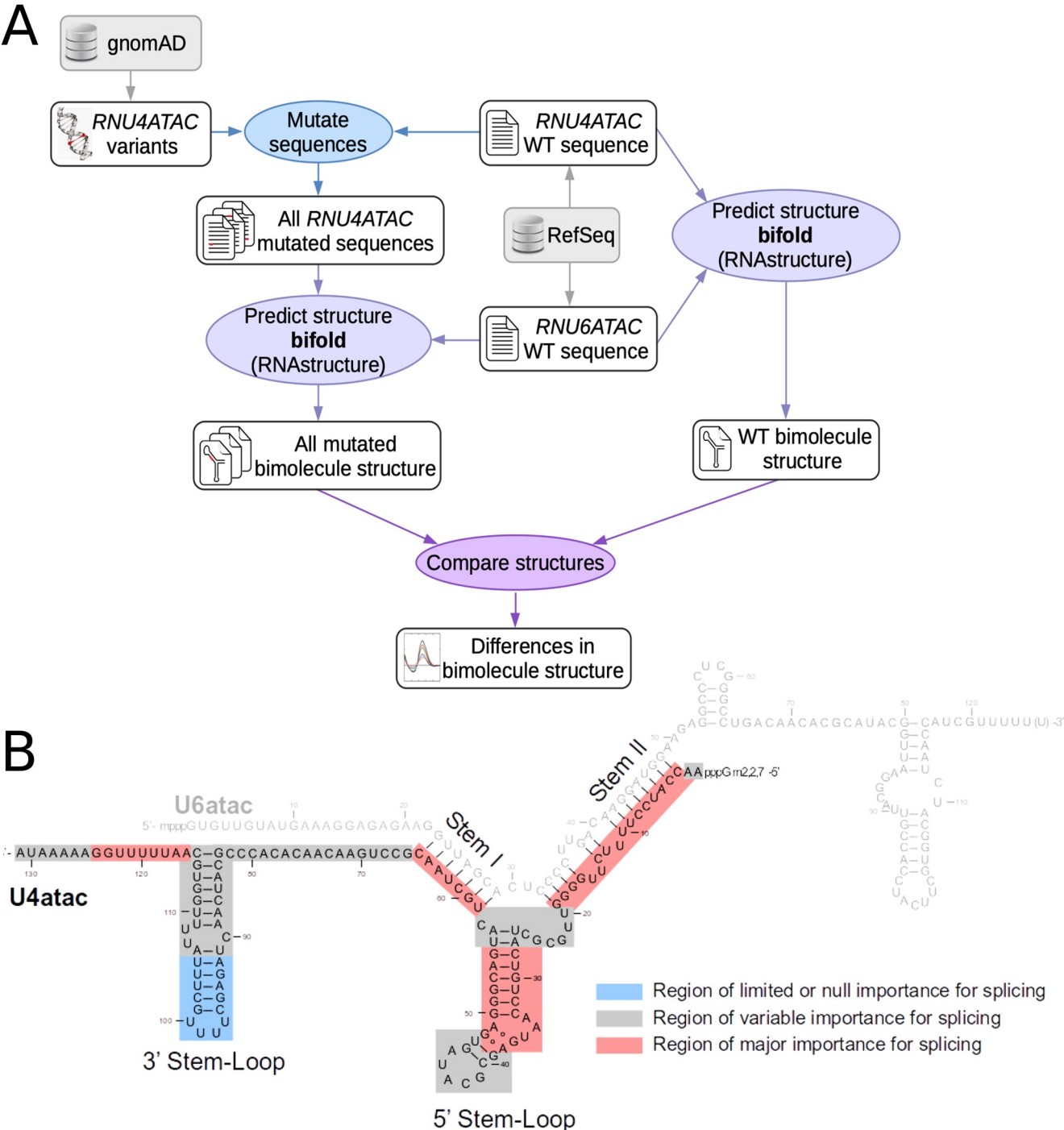

**Fig 2. Prediction of the alteration of the U4atac/U6atac bimolecule bi-dimensional structure. A**. Schematic diagram picturing the bioinformatic pipeline allowing to compute the scores. After generating the mutated *RNU4ATAC* sequences for the 285 variants, the secondary structure of all bimolecules formed by wild type U6atac and mutant U4atac was obtained with the bifold function of the RNAstructure package, compared with that of the wild type bimolecule and differences were scored. **B**. Schematic diagram picturing the importance of U4atac regions for splicing (adapted from [5].) that was used to calculate the score. Score = A x 3 + B x 1 + C x 0.5, where A, B and C are the numbers of modified base pairings in red, grey and blue regions respectively.

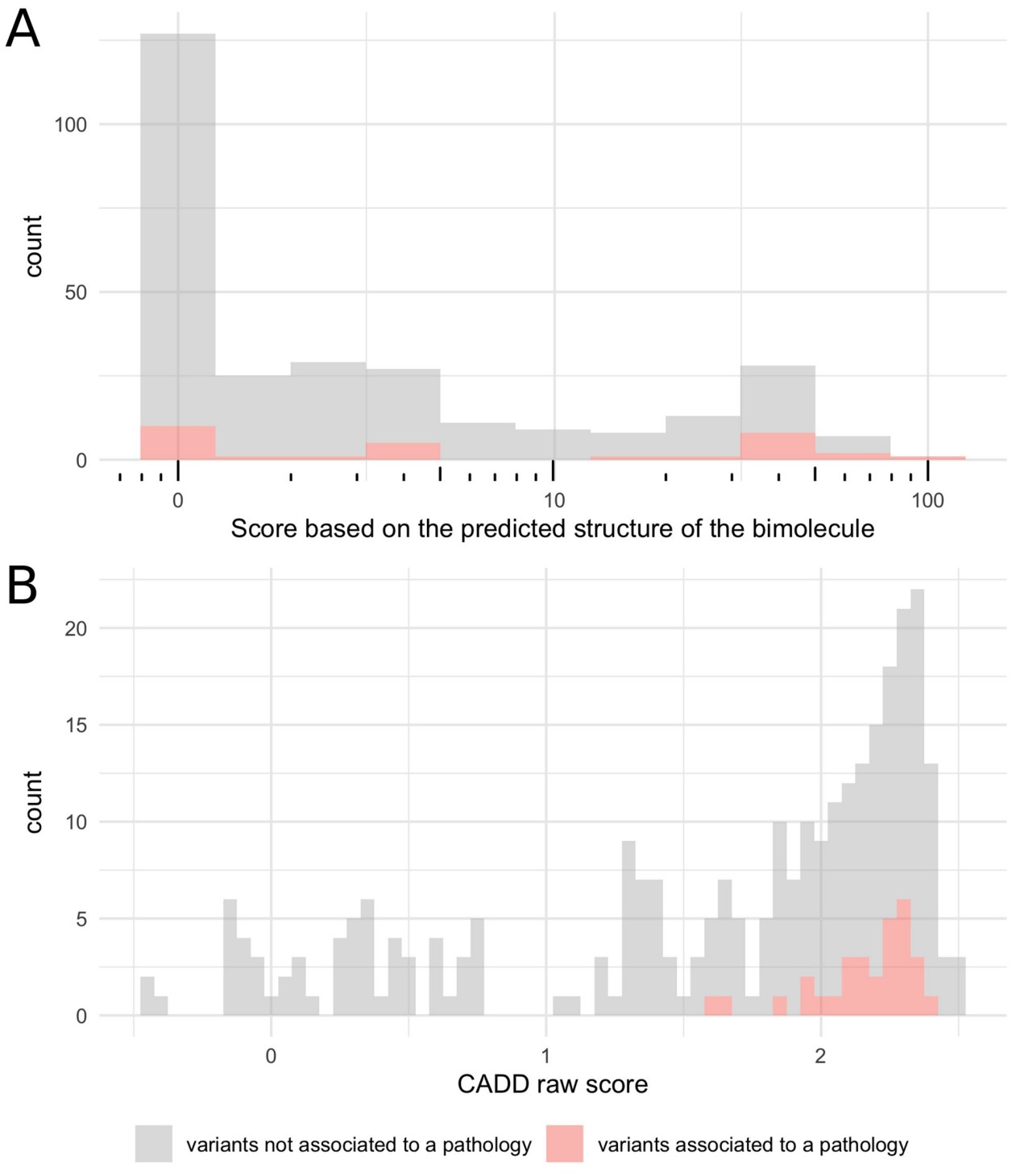

**Fig 3.** Distribution of the scores obtained with A. the bioinformatic pipeline presented in this work and B. the Combined Annotation Dependent Depletion (CADD) prediction tools. CADD combines the results of >60 variant annotation prediction tools into one metric reflecting likelihood of deleteriousness.

-0.44 to 2.52 when considering all the 285 *RNU4ATAC* variants identified (mean: 1.64; median: 1.96), and from 1.60 to 2.39 when considering only the 30 variants identified in patients (S3 Table and Fig 3B). Four variants found in the homozygous state in large-scale sequencing projects have scores inferior to those of the variants associated with a pathology (-0.13 to 1.42), while the other two are within the same ranges (1.66 for n.45A>G and 2.42 for n.58C>T). Overall, the CADD tool appears sensitive for predicting *RNU4ATAC* variant pathogenicity.

## Cellular assay

To set-up a functional assay that could be useful for testing *RNU4ATAC* variants in the diagnostic field, we adapted a system developed to study the functional features of the U6atac and U4atac snRNAs [21] and extensively used thereafter [4,22,29–32]. The original assay is based on the co-transfection in rodent cells (CHO cells) of two minigenes. The first one contains the human *RNU4ATAC* coding region (pUC-U4atac) while the other one contains exons 5–8 as well as introns E, F (an U12-type intron) and G of the human *NOP2* gene, encoding the nucleolar protein P120 (P120 minigene plasmid). To prevent the use of endogenous rodent U4atac molecules, the splicing of the U12-type reporter intron depends in this system on the expression of exogenous U4atac snRNA, achieved through complementary mutations of the sequences of the F intron and of human U4atac. For our cellular model, we aimed to test the functionality of *RNU4ATAC* variants within the context of the native sequence. In order to have reduced interference of the endogenous U4atac, we used primary fibroblasts derived from a TALS patient carrying n.51G>A in the homozygous state. We transiently co-transfected these cells with the native P120 vector and pUC-U4atac constructs containing either the WT or mutated U4atac sequence. The splicing efficiency of the U12-type F intron was then quantified by quantitative RT-PCR (qRT-PCR) (Fig 4A). We thus tested 23 substitutions, among which 7 were found at the heterozygous state in large-scale sequencing projects only and 16 were identified in patients (Fig 4B), as well as the pathogenic 85-nt duplication, each introduced separately in pUC-U4atac. Among the seven variants found in large-scale sequencing projects only, CADD and RNAstructure gave concordant predictions for five and divergent predictions for two of them (S3 Table). Three of these seven variants are localised in the 5' Stem-Loop, one is in Stem I, one is in the single-stranded region between Stem I and the 3' Stem-Loop, and the last two are in the 3' Stem-Loop (Fig 4B).

We first validated our cellular assay by comparing the splicing efficiency of the U12-type reporter intron in control and TALS fibroblasts in the absence or presence of exogenous WT U4atac. We found, as expected, a reduced splicing efficiency in TALS compared to control fibroblasts (~30% versus ~80%). While the splicing efficiency in control cells remained unchanged upon WT *RNU4ATAC* transfection, it was partially restored in TALS fibroblasts, confirming that this cellular model was valuable to test the effect on splicing of *RNU4ATAC* variants (S1A Fig). Of note, transfection of pUC-U4atac in TALS fibroblast cells led to a 6 to 13-fold increase in the amount of U4atac, as shown by qRT-PCR (S1B Fig).

We next compared the splicing efficiency of the U12-type reporter intron following transfection of mutated and WT versions of pUC-U4atac, the splicing efficiency for the WT sequence being set at 100% (see S3 Table and Fig 5). Sixteen of the 17 disease-associated variants led to a splicing efficiency below that obtained with WT U4atac, ranging from that seen in untransfected cells, i.e. ~37% (n.16_100dup, n.37G>A and n.48G>A), to 88% (n.111G>A). The splicing efficiency for the most frequent pathogenic variant, n.51G>A, which is also associated with the most severe phenotypes, was 71%. These effects, even if they were less dramatic, were consistent with those seen with the original cellular assay for three of these variants,

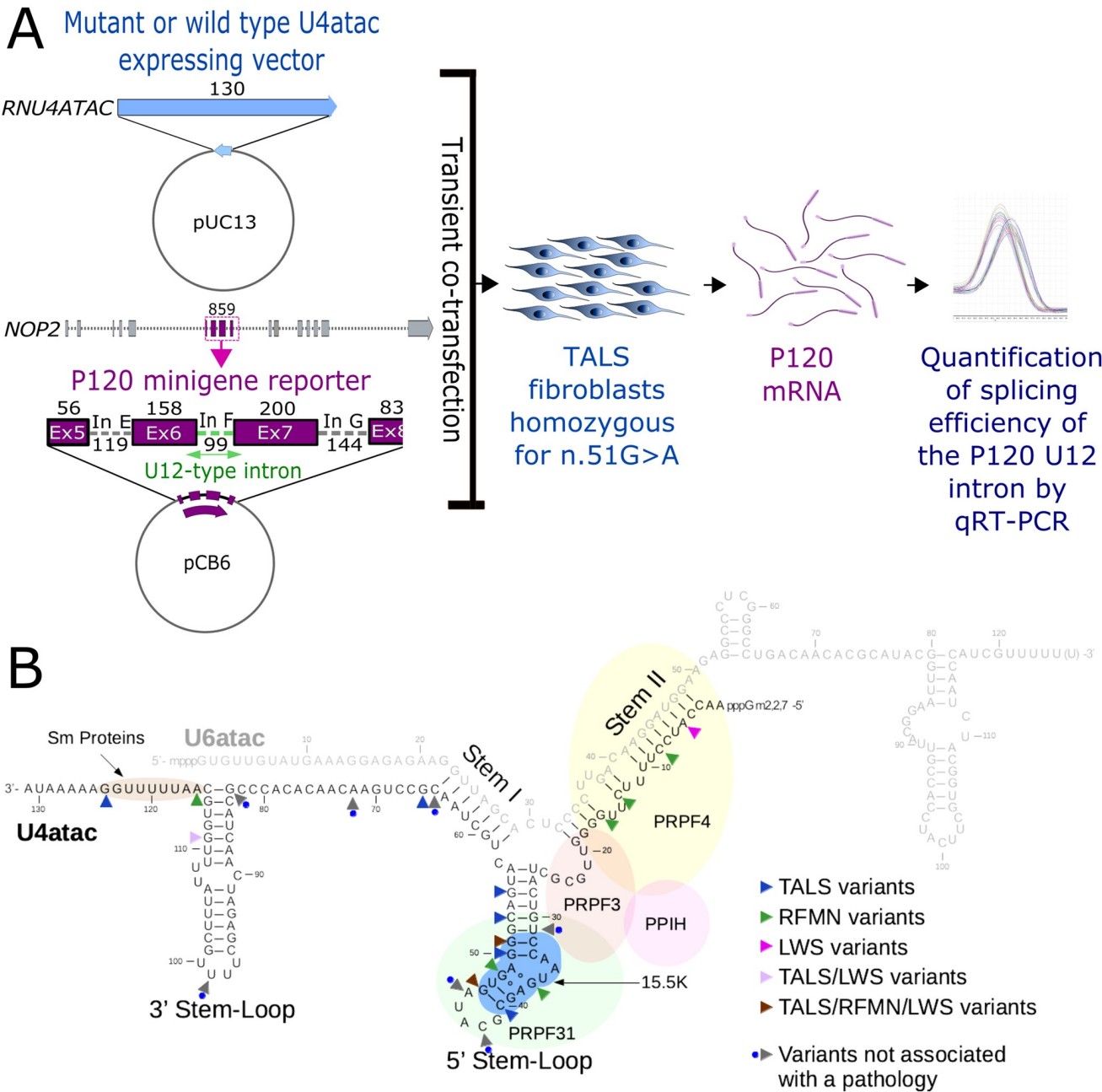

**Fig 4. A**. Schematic representation of the cellular assay. The pUC13 vector containing the mutant or wild type *RNU4ATAC* sequence and the PCB6 plasmid containing the P120 minigene reporter are transiently co-transfected into fibroblast cells derived from a TALS patient homozygous for n.51G>A. The P120 minigene reporter consists of exons 5–8 and introns E, F (U12-type intron) and G of the human *NOP2* gene (*NOP2 Nucleolar protein*). Sizes are given in base pairs. The relative splicing efficiency of the F intron was determined by quantitative RT-PCR. **B**. Localisation of the 23 substitutions tested in the cellular assay. The *RNU4ATAC* genomic sequence is 130 nuceotide-long while the U4atac snRNA is 131 nucleotide-long due to a post-transcriptionally added adenylic acid residue on its 3′-end.

n.51G>A, n.55G>A and n.111G>A [4]. If we consider the diseases in which the studied variants are involved, variants found in RFMN patients were globally associated with stronger effects than those identified in TALS patients, while those identified in LWS had the lowest effects (Fig 5A). On the other hand, none of the seven variants identified only in large-scale

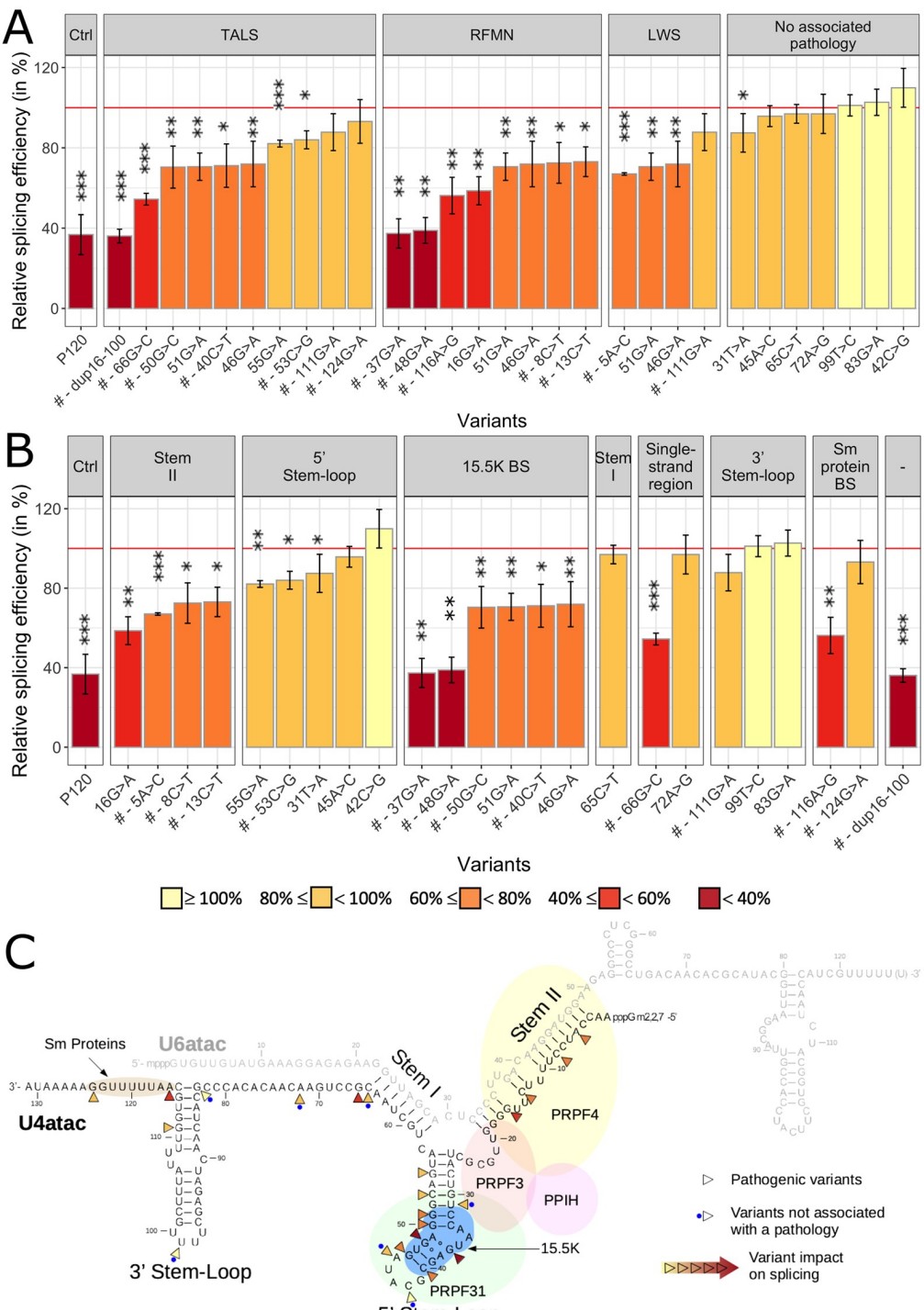

**Fig 5. Results of the cellular assay obtained with the 24 tested variants.** Variants are organised by **A**. associated pathologies (the same variant may appear several times when found in more than one syndrome) or **B**. mutated region of the U4atac/U6atac bi-molecule. Variants are further organised by relative splicing efficiency. The strength of the effect on splicing is colour-coded using a gradient from yellow showing the lowest effect (high splicing efficiency) to dark red showing the highest effect (low splicing efficiency). **C**. Representation of the magnitude of the impact of variants on splicing on the U4atac/U6atac bi-molecule using the same colour code. For clarity, variants' names were simplified; those identified in the compound heterozygous state only are shown with a # (all the others have been identified in both homozygous and compound heterozygous states). Error-bars represent standard error of the mean (SEM) of at least three independent experiments. Statistically significant differences in splicing efficiency are indicated

by asterisks: * *P-values* < 0.05; ** *P-values* < 0.01 and *** *P-values* < 0.001 (one-tailed t-test). The red horizontal line indicates 100% splicing efficiency (i.e. splicing efficiency when transfecting WT *RNU4ATAC*).

sequencing projects impacted the splicing efficiency. When considering the location, the stronger effects were found for the large duplication and for variants residing in the 15.5K protein binding site of the 5' Stem-Loop, in the internal single-stranded region just 3' to Stem I, in the Sm protein-binding site just 3' to the 3' Stem-Loop, followed by variants found in Stem II, whose structure allows the binding of PRPF4 (Fig 5B and 5C). Variants located in the first stem of the 3' and 5' Stem-Loops, outside the 15.5K binding site, have more moderate effects. The only pathogenic variant associated with a full splicing efficiency is n.124G>A, localised in the Sm protein-binding site. The variants identified only in large-scale sequencing projects and without effect in our cellular assay are located in the loops of the 5' and 3' Stem-Loops, in Stem I, in the single-stranded region just 5' to the 3' Stem-Loop or in the first stem of the 5' Stem-Loop.

Because half of the families with a *RNU4ATAC*-associated pathology carry compound heterozygous variants, notably LWS and nearly all RFMN families, we also tested the effect of co-transfecting a combination of two *RNU4ATAC* minigenes carrying variants, thus allowing us to compare the splicing efficiency of variants mimicking three RFMN and seven TALS genotypes (S2 Fig). This resulted in the confirmation that p120 U12 intron splicing deficiency was uncorrelated with disease severity. In particular, the most severe presentation, n.51G>A homozygosity, produced average splicing efficiency values.

Lastly, we investigated the concordance of bioinformatic predictions and cellular assay results by plotting P120 U12 intron splicing efficiency against the scores obtained for each tested variant based on RNAstructure or with CADD. No strong correlation was observed for the former ($r^2$ = 0.006, Fig 6A), while the correlation was better for the latter ($r^2$ = 0.303, Fig 6B).

## Discussion

Non-coding RNAs, whose large number was revealed by comprehensive transcriptomic studies in the recent years, are generally categorised by their size: long non-coding RNAs of up to hundreds of nucleotides, whose function is still poorly characterized, and short RNAs ranging from 19 to 140 nt. These short RNAs comprise the long- and well-known snRNAs, rRNAs and tRNAs, whose function in splicing for the former and in translation for the two latter is essential. Among the few non-coding genes that have presently been found responsible for Mendelian genetic diseases, two are transcribed in snRNA components of the minor spliceosome, namely *RNU4ATAC* and *RNU12*. While most snRNAs, rRNAs and tRNAs genes are present in the human genome in multiple copies, a genomic configuration that protects cells against the damaging effect of mutations, *RNU4ATAC* and *RNU12* are both single copy-genes. *RNU12* biallelic variants have been found so far in a single large consanguineous family in which members displayed autosomal recessively inherited early onset cerebellar ataxia [33], while *RNU4ATAC* biallelic variants have been reported in 46 families to date. Some of these were identified by diagnostic laboratories that did not have any specific expertise of this gene, a situation expected to arise more and more frequently due to the generalisation of diagnostic next-generation sequencing platforms. Yet, interpreting *RNU4ATAC* genetic variants is challenging as classical guidelines do not apply in their integrality.

The difficulties in variant interpretation start from the initial and crucial step of describing the identified variant for laboratories relying for nomenclature on variant-calling pipelines that use reference genome annotation. Most non-coding RNA genes are annotated by aligning

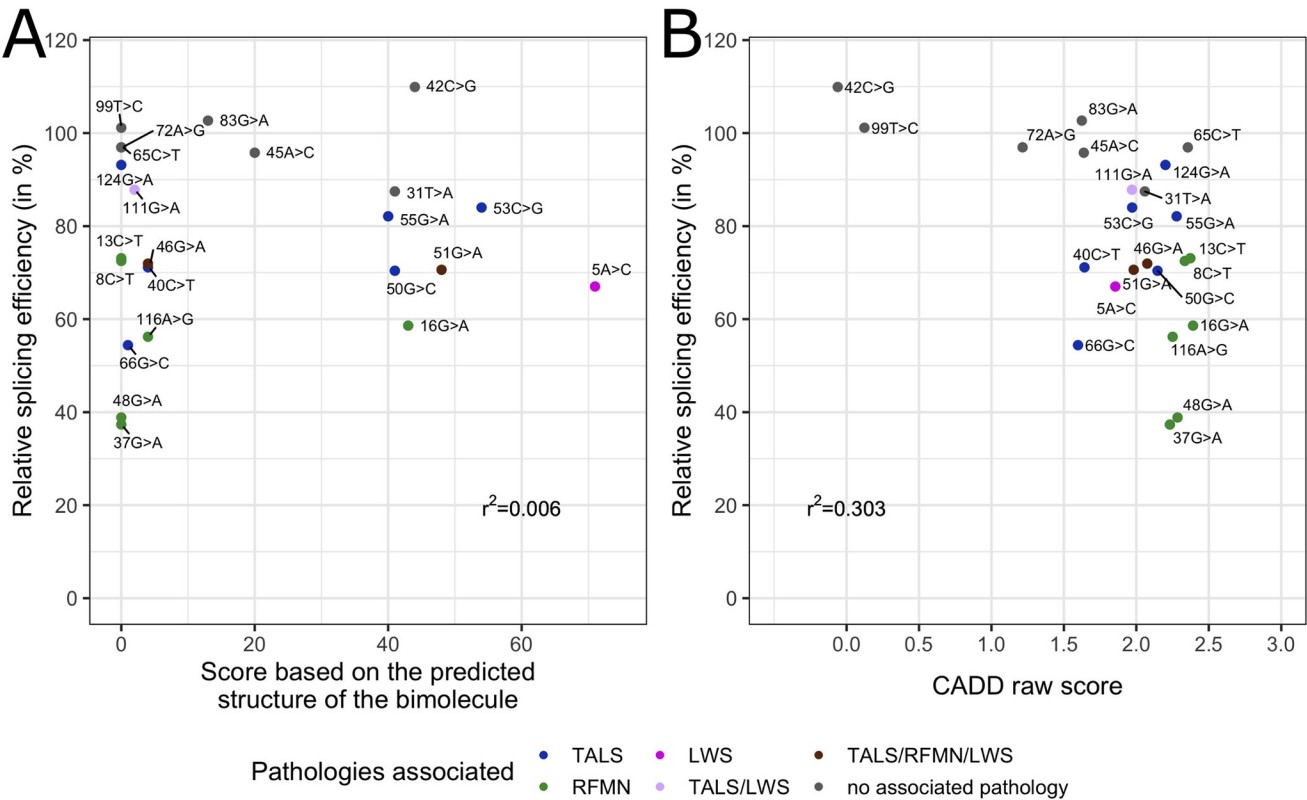

**Fig 6.** Splicing efficiency of the U12-type reporter intron measured after transfection of *RNU4ATAC* variants as a function of **A**. the score obtained with bioinformatic predictions of the U4atac/U6atac bimolecule structure or **B**. the Combined Annotation Dependent Depletion (CADD) raw score. For clarity, variants' names were simplified.

genomic sequences against entries of the Rfam database (the database listing RNA families), thus numbering starts from the first nucleotide of the RNA. Yet for unknown reasons, Homo sapiens *RNU4ATAC* nucleotidic coordinates are inconsistent and erroneous in Rfam, Ensembl, and gnomAD websites with a +1 or +2 shifting [chr2:122,288,457–122,288,583 or chr2:122,288,458–122,288,584 instead of chr2:122,288,456–122,288,585 (Genome build GRCh37/hg19); chr2:121,530,881–121,531,007 instead of chr2:121,530,880–121,531,009 (GRCh38/hg38)]. These small errors are substantial as they induce sequence variant description mistakes. Furthermore, they generate a -1 shifting in the nomenclature of the inferred consequences of all of the *RNU4ATAC* variants listed in gnomAD (for example, Variant ID "2-122288506-G-A", rs188343279, is indicated as n.50G>A when it should be n.51G>A), which can lead to confusion or error when taking into account variant frequency for clinical interpretation.

The knowledge of population genetic variation at a specific locus increases our ability to understand the locus contributions to health and disease. Allele frequency filtering, which rests on established frequency cutoffs (i.e. thresholds above which it is estimated that the allele frequency is incompatible with pathogenicity), are very helpful for appropriate genetic counselling as it allows to remove variants from consideration. The conservative cutoff value of 1% is classically recommended for rare recessive diseases whose prevalence is ill-defined and for which genetic and allelic heterogeneity as well as penetrance are mostly unknown. This filtering step appears of little use in the case of *RNU4ATAC* as the highest overall allelic frequency found among the 282 variants reported in gnomAD (0.29%) is well below this cutoff, not

surprisingly given the high sequence and secondary structure constraints on small RNAs. When looking more closely, three variants have allele frequencies > 1% into one sub-population (n.58C>T, n.87C>T and n.93G>A in Africans, in South Asians or in Ashkenazy Jewishs respectively) and have been identified in the homozygous state in large-scale sequencing projects. They should therefore be considered unlikely to be disease-causing.

Evaluating whether the clinical phenotype of the carrier of a putative pathogenic variant is compatible with the disease associated with the gene is a major determinant of variant pathogenicity. However, it is frequent that molecular genetic diagnosis, in the era of exome and genome sequencing, uncovers previously unsuspected clinical heterogeneity. In the case of *RNU4ATAC*-associated diseases, the patients originally diagnosed with TALS had a highly homogeneous phenotypic presentation. They were found to carry the same homozygous mutation, n.51G>A, or mutations in its close vicinity in the 5' Stem-Loop. However, the identification of new *RNU4ATAC* variants revealed more diverse associated phenotypes in terms of severity, both within the TALS spectrum [34,35] and with the discovery of new *RNU4A-TAC*-associated syndromes, RFMN and LWS. In this context, new *RNU4ATAC* variants should not be dismissed because the phenotype of the carrier(s) does not strictly fit with previous descriptions. In the same line of thought, it should be noted that the delineation of the *RNU4ATAC*-associated syndromes will probably need to be reconsidered based on *RNU4A-TAC* genotypes following the identification of a growing number of *RNU4ATAC* biallelic carriers.

We used two approaches to assess the consequences of *RNU4ATAC* variants, bioinformatics predictions and functional testing. Firstly, we investigated the capacity of U4atac/U6atac structural predictions, using the bifold function of the RNAstructure bioinformatics tool, to predict pathogenicity of *RNU4ATAC* variants. Indeed this tool, which is used in some but not in all publications reporting pathogenic biallelic *RNU4ATAC* variants, had not been evaluated yet regarding its utility for variant classification. We found that ten of the 30 variants identified in patients are not predicted to have structural consequences, not only the six variants located in the Sm protein-binding site, a single-stranded region, but also two variants located in Stem II and two others in the 5' Stem-Loop. This approach, therefore, shows a lack of sensitivity reflecting the fact that the structure of the U4atac-U6atac bimolecule is not the only parameter governing the stability and the function of U4atac. In consequence, this structure prediction tool may be helpful to understand why a variant is pathogenic, but cannot be used on its own to predict pathogenicity. On the other hand, the CADD tool, a widely used integrative annotation built from more than 60 genomic features, appears quite effective at predicting pathogenic variants, as all 30 variants identified in patients had high scores. Concerning the specificity of these two bioinformatic tools, it is difficult to assess, given that there are only three variants, n.58C>T, n.87C>T and n.93G>A that can be assumed to be non pathogenic on the basis of their allelic frequency > 1% in some sub-populations and their detection in the homozygous state in large-scale sequencing studies. Concerning the other three variants found in the homozygous state, n.45A>G, n.91dupT and n.110delT, it may be unwise to exclude their deleteriousness given that only one or two homozygotes were identified respectively among the ~81.000 or ~65.000 screened individuals, and the possibility that some of the gnomAD participants do actually suffer from a disease (although it can be assumed that these variants in the homozygous state are not associated with the most severe TALS phenotype). These three probably non pathogenic variants have null scores with RNAstructure but one of them, n.45A>G, has a high CADD score, suggesting high specificity for the former and a lesser one for the latter. It is too premature at this stage to determine if the fact that nearly two thirds of the variants identified to date have a score in the same range as the pathogenic ones reflects a lack of specificity of this tool or a very high gene constraint.

Secondly, we developed a functional cellular test that assays the splicing activity of mutant U4atac snRNA on a U12-type test intron, the 99 bp-long intron F of the *NOP2* gene encoding the human nucleolar protein P120, in fibroblasts from a TALS patient homozygous for the n.51G>A mutation. The *NOP2* minor intron appeared a suitable test intron as we found that its splicing is systematically impaired, albeit to a small extent, in fibroblasts derived from TALS patients contrary to other less sensitive minor introns (http://lbbe-shiny.univ-lyon1.fr/TALS-RNAseq/) [9]. We found that all but one variant identified in patients were associated with lower splicing efficiency than that obtained with the transfected WT U4atac snRNA, the exception being a variant in the Sm protein-binding site, n.124G>A. Binding of the Sm protein complex has been shown to be an essential step in the snRNP assembly process and is required for stability of the snRNA. Indeed, n.124G>A leads to a large reduction in the amount of U4atac snRNA in transfected CHO cells [12], but this large reduction is possibly counteracted in our system by the important increase in U4atac levels (6–13 times) following transfection of *RNU4ATAC* plasmids (S1 Fig). Interestingly, we noted that the extent of minor splicing impairment of the test intron in our assay did not reflect the severity of the disease observed for the carriers of the tested variants, not surprisingly given the various degrees of retention of the hundreds of minor introns we were able to record in cells derived from TALS patients [9]. On the other hand, the seven *RNU4ATAC* variants never associated with a phenotype did not impact the splicing efficiency in our functional test, even those four predicted by the RNAstructure software to have a strong impact in regions important for splicing, among which three had also a high CADD score. This absence of effect could derive, again, from the fact that we test the splicing of only one U12-type intron while we showed that the level of minor intron retention of the 699 human genes containing a U12-type intron varies depending on the gene, the cell-type and the *RNU4ATAC* genotype of TALS patients [9]. Therefore, a negative result obtained with our cellular assay may not necessarily mean that the tested variant has no effect on minor splicing. Adding up to the previously discussed limitations of U4atac/U6atac secondary structure predictions, the poor concordance of these predictions and cellular test results could also be due to the fact that an important parameter, i.e. the stability of U4atac/U6atac di-snRNP, is not taken into account by the RNAstructure software. Stability is nevertheless a very important factor, certainly explaining the strong effect of the n.66G>C variant located just 3' to Stem I. Indeed, the presence of this variant lengthens and therefore stabilises Stem I, which most probably impairs the U4atac ejection needed for the activation of U6atac [36].

In this work, we 1) provided all the available relevant information needed for *RNU4ATAC* sequence variant classification in clinical molecular diagnostic, 2) evaluated for the first time the utility in the clinical setting of a structural and of the CADD prediction tools, and 3) presented a cellular assay that functionally assesses the variants identified in patients. This functional test allows to move variants from the "probably pathogenic" to the "pathogenic" class, leading to greater confidence in patient reporting and clinical management. Taken together, these data and tools will benefit to diagnostic laboratories in order to insure genetic counselling consistency.

## Supporting information

**S1 Fig. Cellular assay validation. A**. Splicing efficiency of the U12-type reporter intron in control or TALS fibroblasts non transfected (-) or transfected (+) with the P120 and WT pUC-U4atac plasmids, as indicated. Error-bars represent standard error of the mean (SEM) of at least three independent experiments. Statistically significant differences in splicing efficiency are indicated by an asterisk (*P-values* < 0.05). PCR products amplified from unspliced and

spliced transcripts are shown. **B**. Relative amount of U4atac snRNA in TALS fibroblasts non transfected (-) or transfected (+) with the P120 and WT pUC-U4atac plasmids, as indicated. (TIF)

**S2 Fig. Results of the cellular assay obtained after transfection of a single variant or cotransfection of two variants mimicking respectively the homozygous or compound heterozygous genotypes of RFMN and TALS patients.** Error-bars represent standard error of the mean (SEM) of at least three independent experiments. Statistically significant differences in splicing efficiency are indicated by asterisks: * *P-values* < 0.05; ** *P-values* < 0.01 and *** *P-values* < 0.001 (one-tailed t-test). The red horizontal line indicates 100% splicing efficiency. (TIF)

**S1 Table. Census of all patients with biallelic *RNU4ATAC* pathogenic variants published in the literature (May 2020).**
(XLSX)

**S2 Table. Census of all *RNU4ATAC* variants present in the gnomAD resource (v2.1, January 2020).**
(XLSX)

**S3 Table. Scores computed based on an RNA structure prediction tool and CADD scores for the variants identified in patients and/or in large-scale sequencing projects; results from the cellular assay for the 24 *RNU4ATAC* variants tested.**
(XLSX)

## Acknowledgments

We thank the CBC Biotec biobank for biosample management (Emilie Chopin, Isabelle Rouvet), and the GENDEV team members for stimulating discussions.

## Author Contributions

**Conceptualization:** Clara Benoit-Pilven, Audrey Putoux, Patrick Edery, Sylvie Mazoyer.

**Data curation:** Clara Benoit-Pilven.

**Formal analysis:** Clara Benoit-Pilven, Alicia Besson.

**Funding acquisition:** Anne-Louise Leutenegger, Vincent Lacroix, Patrick Edery, Sylvie Mazoyer.

**Investigation:** Clara Benoit-Pilven, Alicia Besson, Clément Saccaro, Justine Guguin, Gabriel Sala.

**Methodology:** Clara Benoit-Pilven, Audric Cologne, Marion Delous, Gaetan Lesca, Anne-Louise Leutenegger, Vincent Lacroix, Patrick Edery, Sylvie Mazoyer.

**Resources:** Audrey Putoux, Claire Benetollo, Richard A. Padgett.

**Supervision:** Audrey Putoux, Claire Benetollo, Patrick Edery, Sylvie Mazoyer.

**Writing – original draft:** Clara Benoit-Pilven, Alicia Besson, Sylvie Mazoyer.

**Writing – review & editing:** Clara Benoit-Pilven, Alicia Besson, Audrey Putoux, Claire Benetollo, Clément Saccaro, Justine Guguin, Gabriel Sala, Audric Cologne, Marion Delous, Gaetan Lesca, Richard A. Padgett, Anne-Louise Leutenegger, Vincent Lacroix, Patrick Edery, Sylvie Mazoyer.

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
