## [Decision Letter · Decision Letter 0]

21 Apr 2020

PONE-D-20-06355

Clinical interpretation of variants identified in RNU4ATAC, a non-coding spliceosomal gene

PLOS ONE

Dear Dr. Sylvie Mazoyer,

Thank you for submitting your manuscript to PLOS ONE. After careful consideration, we feel that it has merit but does not fully meet PLOS ONE’s publication criteria as it currently stands. Therefore, we invite you to submit a revised version of the manuscript that addresses the points raised during the review process.

The manuscript is interesting and can be of interest to the scientific community, but serious attention to figures and readability of the manuscript is a necessity. 

Please carefully address the concerns of the reviewers especially pay attention to eventual endogeneous backgground expression of RNU4ATAC.  

We would appreciate receiving your revised manuscript by Jun 05 2020 11:59PM. To enhance the reproducibility of your results, we recommend that if applicable you deposit your laboratory protocols in protocols.io, where a protocol can be assigned its own identifier (DOI) such that it can be cited independently in the future. For instructions see: http://journals.plos.org/plosone/s/submission-guidelines#loc-laboratory-protocols

We look forward to receiving your revised manuscript.

Kind regards,

Klaus Brusgaard

Academic Editor

PLOS ONE

Journal Requirements:

2. Thank you for including the following ethics statement on the submission details page:

' Authorisation for their collection and their use in research has been granted by the Ministry of Research, by the Comité de protection des Personnes Sud-Est IV and the Regional Agency for Hospital Services under the number DC-2015-2566. The project has been approved by the local ethics committee of the Hospices Civils de Lyon.'

Please also include this information in the ethics statement in the Methods section of your manuscript.

'This work was supported by CNRS, Inserm, Université Paris7 and Université Lyon 1 through recurrent funding, the ANR Aster (no. ANR-16-CE23-0001) and U4ATAC-BRAIN (no. ANR-18CE12-0007-01) grants and an Inserm/Hospices Civils de Lyon grant to P.E. (Contrat d'Interface pour Hospitaliers). A.C. was supported by a grant from Inria (Thése Inria-Inserm “Médecine Numérique” - 2016) and C.B.P. by a grant from the Fondation pour la Recherche Médicale to P.E. (Financement d’un ingénieur - ING20160435660).'

'The funders had no role in study design, data collection and analysis, decision to publish, or preparation of the manuscript.'

Please clarify the sources of funding (financial or material support) for your study. List the grants or organizations that supported your study, including funding received from your institution.State what role the funders took in the study. If the funders had no role in your study, please state: “The funders had no role in study design, data collection and analysis, decision to publish, or preparation of the manuscript.”If any authors received a salary from any of your funders, please state which authors and which funders.If you did not receive any funding for this study, please state: “The authors received no specific funding for this work.”Please include your amended statements within your cover letter; we will change the online submission form on your behalf.

5. Your ethics statement must appear in the Methods section of your manuscript. If your ethics statement is written in any section besides the Methods, please move it to the Methods section and delete it from any other section. Please also ensure that your ethics statement is included in your manuscript, as the ethics section of your online submission will not be published alongside your manuscript.

Reviewers' comments:

Reviewer's Responses to Questions

**Comments to the Author**

1. Is the manuscript technically sound, and do the data support the conclusions?

Reviewer #1: Yes

Reviewer #2: Partly

2. Has the statistical analysis been performed appropriately and rigorously? 

Reviewer #1: Yes

Reviewer #2: No

3. Have the authors made all data underlying the findings in their manuscript fully available?

Reviewer #1: No

Reviewer #2: Yes

4. Is the manuscript presented in an intelligible fashion and written in standard English?

Reviewer #1: Yes

Reviewer #2: No

5. Review Comments to the Author

Reviewer #1: This paper looks at a very rare, deadly disease, that is caused by variants in a non-coding spliceosome gene. Variant interpretation is difficult, because existing methods of variant interpretation have focused on coding genes. This paper describes surveying known variants from patients, evaluation of allele frequencies from gnomAD, and splice efficiency assays to evaluate a subset of the variants. The results and approach should be of interested to other non-coding disease gene scenerios.

Figure 1 is impressive -- the vast majority of all variants are in known domains.

Figure 2B -- The authors have a link between "Mutate sequence" and RNU4ATAC WT sequence" -- presumably the flow of data is from "RNU4ATAC WT sequence" is to the "Mutate sequences" step. If that link is not bidirectional, it should have an arrow. The same applies to the other links -- since you are using arrows for the other links.

Figure 4 fonts are difficult to read. Maybe do not use "bold."

Figure 5A/B labels are impossible to read. Figure 5B caption says "Variants are further

organised by relative splicing efficiency, the stronger the effect on splicing is, the more intense the colour." Given that I can't read the labels -- I don't know if red is more intense, or yellow is more intense. How do you define an intense color? The range of effect appears to be <0.4% (red), and >= 1% (yellow). Presumably the yellow is greatest effect (has the highest splicing efficiency) on splicing efficiency.

Am I able to distinguish which variants in Figure 5C correspond to variants in 5A/5B? I can't tell because I can't read the labels.

What does the red horizontal line in Figure 5A/B mean? I can see from Supp Figure 2 that the red horizontal line is at 1% efficiency. Why is the threshold important? Whoever made Supp Figure 2 needs to redo figure 5A/5B.

Supp Table 3.xlsx -> Scores for all variants -> This tab contains the "CADD phredScore."

Supp Table 3.xlsx -> Scores for disease-associated -> This tab only has the "CADD rawScore." I believe readers will be interested in the "CADD phredScore" scores for the "disease-associated" variants in the event CADD scores are updated. Can you add that column to this table?

Figure 3 and Figure 6 legend should really say you used the "raw" CADD scores.

It looks like they assayed 35 cells for splicing efficiency, against 2 cells for controls (Supp Table 3.xlsx -> Results cell.assay tab) -- each cell is a different variant -- performed in triplicate according to the methods.

Reviewer #2: Review: Clinical interpretation of variants identified in RNU4ATAC, a non-coding spliceosomal gene

Summary: In the manuscript by Mazoyer et al, they describe the spectrum of variants reported in the literature and then proceed to bioinformatics and cell-based assessment of variant effects. The prioritization of noncoding variants is an understudied and important area of human genetics, so this study has the potential to add significantly to the field. The authors have modified a cell based assay and use this modified assay to validate computationally identified variants as likely pathogenic. However, there are significant concerns. The cell based assay measures splicing of RNA4ATAC. However, the sensitivity of this assay is not established here.

Major concerns:

- What is the sensitivity of the cell assay? A) The authors have focused on a 'positive' signal (i.e. impaired splicing) in cells transfected with mutant RNA4ATAC RNA. Splcing was demonstrated to be variable (line 370), but this variability was not accounted for in the calculation of splicing efficiency. Further, there was background expression of RNU4ATAC, wildtype or mutant, which could affect some variants more than others if there are any mixed heterodimers forming. Why was an RNU4ATAC null line not used for the assay? Or specific knockdown of endogenous U4atac to ensure that the majority of splicing activity was from the transfected RNA4ATAC and not endogenous activity, as there could be compensatory upregulation of the endogenous gene expression? Similarly, is the effect of heterodimers just due to the increased among of protein expressed in dual-transfected cells (line 396)? Why were those 23 variants selected? That information is necessary to interpret any results. And does this assay have any relevance for downstream function? What percentage decrease would actually harm a cell? The tolerance of the system to decreased splicing efficiency has not been determined in this assay so the disease relevance is limited.

B) False positives: The authors claim that essentially all tested variants from affected individuals perturb splicing. However a number of variants that are common and that should be not be pathogenic should also be tested. Does any variant that perturbs 'secondary structure' perturb RNA splicing? Do any loop variants affect splicing?

- Statistical analysis is needed throughout. Currently the results are largely descriptive. For example, is the difference in age at death truly different for n.51G>A (line 246)? Simple comparison of age or proportions below a certain age is necessary to support this claim. Similarly for the bioinformatic predictions, what is the enrichment for “pathogenic” vs population variants at each score threshold? The proportions must be evaluated statistically and not simply described. The CADD analysis similarly needs burden evaluation. None of Figure 5 has any statistical evaluation – which variants are truly different from reference? Were they different in all 3 replicates? This needs to be re-calculated after protein expression levels are accounted for.

- What is the scientific basis for assaying U4atac/U6atac interaction as a disease-relevant measure of variant effect? This is never substantiated and therefore has limited applicability to human disease.

- Carefully review genomic coordinates before claiming that a reference genome is incorrect. Some resources are 0-based while others are 1-based. The authors should submit firm evidence of their claims that references are truly off by 1 and it is not an expected difference in data structure. This is especially true as much of the discussion is devoted to denouncing the references (line 424-437).

- The bioinformatic approach needs to be more fully described. Why was this RNA structure tool selected? Were alternatives tried? Since the bioinformatic prioritization seems to have not had any correlations, this null result needs to be more fully explored. A description of why the bioinformatic score was selected needs to be included. How successfully and relevant is the scoring system by Merico? Were other approaches tried?

- Selective pressure described in 283 should be substantiated. How low is the frequency compared to variation across other noncoding RNAs? What is the z-score for these variants compared to other noncoding RNA bases? Selective pressure should not be determined by the raw value alone

- The population frequencies described in line 286-297 are highly misleading because the sub-populations mentioned are too small and thus the allele frequency should have a large confidence interval. Given that a large proportion of cases are compound heterozygous genotypes, this also seems unnecessary to dwell on.

-

Minor concerns:

- The text suffers from long sentences with many subordinate clauses. The clarity of the scientific message would benefit from rewriting. This is particularly true of the first and last paragraphs of the introduction, and all descriptions of the cellular assay.

- In all reported cases, can it be confirmed that no other potentially pathogenic variants were identified? More description in the methods as to the process for assessing of the validity of the reported variants is necessary before proceeding to bioinformatic or functional assessment

- It is stated that all data are available, but the VCFs for the large cohorts described in the studies should be clearly available for assessment of other potentially pathogenic variants. Are those available from other publications? If so, that information should be provided in the methods

- It seems that the preferred term is MOPD1 instead of TALS, to prioritize descriptions of pathology rather than named after the describing team

- A better summary of the unique and overlapping features of the three syndromes could be provided as a supplemental table

- Line 68 references unexplained deaths but there seemed to be neurological etiologies described in the other papers

- The third paragraph of the introduction needs far more references to support the scientific statements. It would be helpful to specify which tri-snRNPs are disrupted in MOPD1 (line 94-95)

- Line 123 should not have an apostrophe and gnomAD should not be capitalized (here and throughout)

- Line 133 totalising is not a word

- A specific link to the github mentioned in line 150 should be provided

- Detailed discussion of the domains of U4atac in 222-243 would be better for the introduction than results

- Similarly, much of the variant description in line 251-269 is highly redundant and could be summarized in a much shorter form

- Half as less is not a phrase, in line 275

- Figure 2A needs to be significantly reformatted to be linear. The zig zag structure makes it as confusing as the text

- Figure 3A would be more clear as a log-scale x-axis to minimize the white space

- Figures 4A/5C are very poor quality and cannot be read

- Figure 5 text is too small to be read

- In discussion, noncoding genes are not only transcribed long RNAs or short RNAs. Some encode micropeptides, or have other functions. Therefore the word either in line 408 is not appropriate.

- For Figure 6, are there common features of the variants with high vs low correlation? Does that show that the cellular assay has important information about some of the very low CADD scores? If those two low CADD scores are excluded, what is the new R2 value. Also that figure does not need as many decimal points displayed in the figure.

- Figure 1 there is an extra triangle pointing to the U on the right side of the 5-stem loop. What does this signify? Also the colors are too similar in Figure 1 – especially the purple and dark blue

- Abstract the phrase: "… that allows to measure the splicing efficiency of RNU4ATAC variants on a minor (U12-type) reporter intron" needs modification.

6. PLOS authors have the option to publish the peer review history of their article (what does this mean?). If published, this will include your full peer review and any attached files.

Reviewer #1: No

Reviewer #2: No

---

## [Author Response · Author response to Decision Letter 0]

2 Jun 2020

Reviewer #1: This paper looks at a very rare, deadly disease, that is caused by variants in a non-coding spliceosome gene. Variant interpretation is difficult, because existing methods of variant interpretation have focused on coding genes. This paper describes surveying known variants from patients, evaluation of allele frequencies from gnomAD, and splice efficiency assays to evaluate a subset of the variants. The results and approach should be of interested to other non-coding disease gene scenerios.

Figure 1 is impressive -- the vast majority of all variants are in known domains.

Indeed!

Figure 2B -- The authors have a link between "Mutate sequence" and RNU4ATAC WT sequence" -- presumably the flow of data is from "RNU4ATAC WT sequence" is to the "Mutate sequences" step. If that link is not bidirectional, it should have an arrow. The same applies to the other links -- since you are using arrows for the other links.

We have changed the figure according to the recommendations.

Figure 4 fonts are difficult to read. Maybe do not use "bold."

We apologize for the poor quality of the figure. We improved it.

Figure 5A/B labels are impossible to read. Figure 5B caption says "Variants are further organised by relative splicing efficiency, the stronger the effect on splicing is, the more intense the colour." Given that I can't read the labels -- I don't know if red is more intense, or yellow is more intense. How do you define an intense color? The range of effect appears to be <0.4% (red), and >= 1% (yellow). Presumably the yellow is greatest effect (has the highest splicing efficiency) on splicing efficiency.

We apologize for the lack of clarity of the legend and labels in Figure 5A/B. We increased the size of the labels and rephrased the legend: “The strength of the effect on splicing is colour-coded using a gradient from yellow showing the lowest effect (high splicing efficiency) to dark red showing the highest effect (low splicing efficiency).” (lines 823-825)

Am I able to distinguish which variants in Figure 5C correspond to variants in 5A/5B? I can't tell because I can't read the labels.

We increased the size of the labels in Figure 5 to improve its readability. 

What does the red horizontal line in Figure 5A/B mean? I can see from Supp Figure 2 that the red horizontal line is at 1% efficiency. Why is the threshold important? Whoever made Supp Figure 2 needs to redo figure 5A/5B.

The red horizontal line in Figure 5A/B and Supp Figure 2, at 100% efficiency, represents splicing efficiency when transfecting wild-type RNU4ATAC sequence. This information was added in the figures’ legends (lines 832-833).

Figure 5A/B and Supp Figure 2 were harmonised. 

Supp Table 3.xlsx -> Scores for all variants -> This tab contains the "CADD phredScore."

Supp Table 3.xlsx -> Scores for disease-associated -> This tab only has the "CADD rawScore." I believe readers will be interested in the "CADD phredScore" scores for the "disease-associated" variants in the event CADD scores are updated. Can you add that column to this table?

We added the CADD phredScore in the 3 tabs that did not contain it, i.e. “Scores for disease-associated”, “Scores gnomAD hmz variants” and “Scores variants cell. assay”.

Figure 3 and Figure 6 legend should really say you used the "raw" CADD scores.

We changed the label “CADD score” in Figure 3 and Figure 6 to “CADD raw score” as suggested. 

It looks like they assayed 35 cells for splicing efficiency, against 2 cells for controls (Supp Table 3.xlsx -> Results cell.assay tab) -- each cell is a different variant -- performed in triplicate according to the methods.

This is right, control fibroblasts were not used for our assay as it turned out that transfecting these cells with WT RNU4ATAC did not increase splicing efficiency of our test minor intron, contrary to what happened in patient fibroblasts. We thus used only fibroblasts derived from a TALS patient.

Reviewer #2: Review: Clinical interpretation of variants identified in RNU4ATAC, a non-coding spliceosomal gene

Summary: In the manuscript by Mazoyer et al, they describe the spectrum of variants reported in the literature and then proceed to bioinformatics and cell-based assessment of variant effects. The prioritization of noncoding variants is an understudied and important area of human genetics, so this study has the potential to add significantly to the field. The authors have modified a cell based assay and use this modified assay to validate computationally identified variants as likely pathogenic. However, there are significant concerns. The cell based assay measures splicing of RNA4ATAC. However, the sensitivity of this assay is not established here.

Major concerns:

- What is the sensitivity of the cell assay? 

A) The authors have focused on a 'positive' signal (i.e. impaired splicing) in cells transfected with mutant RNA4ATAC RNA. Splcing was demonstrated to be variable (line 370), but this variability was not accounted for in the calculation of splicing efficiency. 

Indeed, we found that the increase in the amount of U4atac following transfection of pUC-U4atac in TALS fibroblast cells was variable from one transfection to the other (6 to 13-fold increase), most probably due to the variability of transfection efficiency. We circumvent this well-known technical bias by calculating the relative splicing efficiency of a variant to that of the WT of the same experiment, as explained in Material and Methods (lines 258-263). To make it clearer, we added the information that “Every transfection experiment performed to test a batch of RNU4ATAC variants included a set of cells transfected with the WT U4atac snRNA expression plasmid.“ (lines 225-227).

Further, there was background expression of RNU4ATAC, wildtype or mutant, which could affect some variants more than others if there are any mixed heterodimers forming. 

There is no background expression of wildtype RNU4ATAC, as the transfected cells are homozygous for the n.51G>A variant. The level of the endogenous mutated U4atac snRNA being much lower than that of the transfected U4atac (see Fig S1B), we do not believe that it could have an effect. Further, the possibility of bias introduced by "mixed heterodimers" is null as U4atac does not homodimerize. Heterodimers form only between U4atac and U6atac.

Why was an RNU4ATAC null line not used for the assay? 

At the time of genome editing, the question of a null cell line is reasonable. Nevertheless, given the essential role of U4atac in splicing, we assumed that its complete loss of function would impede cell survival, preventing us to perform experiments. For this reason, we rather chose to use a hypomorphic mutation (n.51G>A) allowing cell survival, and took the opportunity of patient cells we had the chance to collect.

In favour of the assumption that a null cell line would not grow, we generated u4atac KO zebrafish line, and observed that mutant embryos do not survive beyond 24 hpf, exhibiting a growth arrest followed by necrosis soon after the consumption of maternally contributed spliced transcripts (unpublished results). 

Or specific knockdown of endogenous U4atac to ensure that the majority of splicing activity was from the transfected RNA4ATAC and not endogenous activity, as there could be compensatory upregulation of the endogenous gene expression? 

Specific knockdown of endogenous U4atac to ensure that the majority of splicing activity came from the transfected RNU4ATAC was not necessary as the endogenous activity of U4atac carrying the n.51G>A variant (the only molecule present in these cells) is low, as shown in Figure S1. Even if a compensatory upregulation of the endogenous gene expression happens, it still does not allow to splice efficiently the transfected P120 test intron. Upon transfection of WT U4atac, we see a 2 fold increase in splicing efficiency of the P120 transcript minor intron.

Similarly, is the effect of heterodimers just due to the increased among of protein expressed in dual-transfected cells (line 396)? 

In order to test the joint effect of two different variants, co-transfections of half the amount of each version of U4atac snRNA expression plasmids, as compared to single transfection, was realised (lines 223-225).

Why were those 23 variants selected? That information is necessary to interpret any results. 

We tested more than half of the variants identified in patients anywhere in the world (17 out of 30); they were chosen on the basis of their chronological identification and of their localisation, to make sure that all regions were represented. The seven variants not associated (as yet) with a disease (found in large-scale sequencing projects) were chosen on the basis of their predicted impact on the bimolecule according to CADD and RNAstructure, and of their localisation, once again to make sure that all the important regions were represented (lines 424-426). 

And does this assay have any relevance for downstream function? 

U4atac’s only known function is minor splicing. Impaired minor splicing efficiency of all transcripts containing U12 introns has been evidenced in RNA-seq analyses in cells from patients for TALS and RFMN syndromes (Merico et al 2015; Dinur Schejter et al 2017; Heremans et al 2018; Cologne et al 2019) (This information was added in lines 117-118). In particular, we found in our analysis (Cologne et al 2019) that the NOP2 minor intron splicing was systematically impaired in fibroblasts derived from 5 TALS patients, as compared to controls (lines 587-590).

What percentage decrease would actually harm a cell? 

This is a very interesting question that we are tackling in the laboratory, but this is altogether another project. 

The tolerance of the system to decreased splicing efficiency has not been determined in this assay so the disease relevance is limited.

Our assay doesn’t measure and doesn’t depend on cell viability and/or functioning capacity. Its goal is to measure splicing efficiency and no more. The role of minor splicing impairment in the etiology of diseases due to RNU4ATAC bi-allelic variants is certain (see the answer to the previous question).

B) False positives: The authors claim that essentially all tested variants from affected individuals perturb splicing. However a number of variants that are common and that should be not be pathogenic should also be tested. 

There are no RNU4ATAC common variants (minor allele frequency > 5%) and no RNU4ATAC polymorphism (minor allele frequency > 1%). We tested 7 variants that had never been found in patients (minor allele frequency ranging from 0.01% to 0.06%) and found that six of them didn’t impact splicing efficiency of our test minor intron. The remaining one, n.31T>A, associated with a slight splicing efficiency decrease, is located at the frontier of the 15.5K protein binding-site.

Does any variant that perturbs 'secondary structure' perturb RNA splicing? 

We showed in our analysis that variants predicted to perturb U4atac/U6atac secondary structure do not always perturb P120 minor intron splicing (see Fig6A). However, we cannot be sure that these variants indeed perturb the secondary structure and even if they do, their effect on splicing efficiency is likely to depend on their location in the U4atac molecule. 

Do any loop variants affect splicing?

We tested three “loop variants” (n.42C>G, n.45A>C and n.99T>C), none had an effect on splicing efficiency. 

- Statistical analysis is needed throughout. Currently the results are largely descriptive. For example, is the difference in age at death truly different for n.51G>A (line 246)? Simple comparison of age or proportions below a certain age is necessary to support this claim.

The difference in age at death of 17 TALS cases was noted as early as in 2012, one year after the publication of RNU4ATAC being the TALS gene. Indeed, Nagy et al. found a significant difference in survival between those carrying two copies of the 51G>A mutation (mean survival=10.4 months) vs those with zero or one copies of the 51G>A mutation (mean survival 78.75 months) (p value=0.02 using the log-rank test for differences in survival curves) (2012). 

Here is what the survival curve looks like when taking into account the 22 n.51G>A homozygous vs the 16 TALS patients with other genotypes published to date. 

 Similarly for the bioinformatic predictions, what is the enrichment for “pathogenic” vs population variants at each score threshold? The proportions must be evaluated statistically and not simply described. 

We didn’t provide statistical analyses because we think that these proportions are only indicative as we can’t consider the variants found in large-scale sequencing studies as “non pathogenic”. Indeed, the large majority of variants found at the heterozygous state in population studies could potentially lead to a disease if associated in trans with another variant or if present at the homozygous state. There are only three variants that can be assumed to be non pathogenic on the basis of their allelic frequency > 1% in some sub-populations and their detection in the homozygous state in large-scale sequencing studies (n.58C>T, n.87C>T and n.93G>A).

The CADD analysis similarly needs burden evaluation. 

The answer is the same as for structural conformation predictions.

None of Figure 5 has any statistical evaluation – which variants are truly different from reference? Were they different in all 3 replicates? This needs to be re-calculated after protein expression levels are accounted for.

Differences between replicates are shown with the error bars. As explained earlier, transfection efficiency differences are taken into account. 

Statistically significant differences in splicing efficiency are now indicated by asterisks (* P-values<0.05; ** P-values<0.01 and *** P-values<0.001).

- What is the scientific basis for assaying U4atac/U6atac interaction as a disease-relevant measure of variant effect? This is never substantiated and therefore has limited applicability to human disease.

We clarified in the text the importance of the U4atac/U6atac bimolecule for minor splicing by adding more information in the introduction (lines 110-115).

« The U4atac/U6atac small nuclear ribonucleoprotein particle (U4atac/U6atac di-snRNP) is composed of U4atac snRNA stably base-paired with U6atac snRNA and of seven Sm proteins and other particle-specific proteins. This di-snRNP then associates with the U5 snRNP, forming the U4atac-U6atac.U5 tri-snRNPS, a component of the pre-catalytic complex which gains its catalytic activity that will allow to excise the intron following the dissociation of U4atac and the pairing of U6atac with U12. Several RNU4ATAC mutations identified in TALS patients have been shown to result in defects in minor tri-snRNP formation [11]. Further, transcriptomic analyses of cells from RFMN and TALS patients revealed massive U12 intron retentions [5, 9, 36, 37]. »

- Carefully review genomic coordinates before claiming that a reference genome is incorrect. Some resources are 0-based while others are 1-based. The authors should submit firm evidence of their claims that references are truly off by 1 and it is not an expected difference in data structure. This is especially true as much of the discussion is devoted to denouncing the references (line 424-437).

We are well aware that some resources are 0-based while others are 1-based. However, whatever the system used, the HGVS variant nomenclature should be respected, which is not the case for RNU4ATAC variants described in gnomAD (n.51G>A is reported as n.50G>A). We can thus rightfully say that there is indeed an error in RNU4ATAC genomic coordinates in some databases. We signaled this to gnomAD and Rfam.

- The bioinformatic approach needs to be more fully described. Why was this RNA structure tool selected? Were alternatives tried? 

Presently, there are only two bioinformatic tools that predict secondary structure of bimolecules of RNAs. RNAstructure is the only tool that can be used at the moment, as the other one proposed by Vienna RNA gives a U4atac/U6atac structure different from the published one. This has been added in the text (line 369).

Since the bioinformatic prioritization seems to have not had any correlations, this null result needs to be more fully explored. A description of why the bioinformatic score was selected needs to be included. How successfully and relevant is the scoring system by Merico? Were other approaches tried?

The problem with the bioinformatic prioritization based on structural predictions comes from the fact that ten of the 30 variants identified in patients are not predicted to have structural consequences according to RNAstructure, as we discuss in the manuscript (lines 559-566). A different scoring system would provide similar results. We would have liked to add, on the structural predictions, data concerning for example the nucleotides involved in protein binding but the available information is not precise enough.

- Selective pressure described in 283 should be substantiated. How low is the frequency compared to variation across other noncoding RNAs? 

The cited sentence reads as follows: “The most frequent variant identified, n.23C>T (never identified in patients), is present in only 0.29% of the screened alleles, suggesting a strong selective pressure against variations in this gene”. We do not say that selective pressure is more important for RNU4ATAC than for other short noncoding genes. 

What is the z-score for these variants compared to other noncoding RNA bases? Selective pressure should not be determined by the raw value alone

We believe that this type of analysis falls beyond the scope of our publication.

- The population frequencies described in line 286-297 are highly misleading because the sub-populations mentioned are too small and thus the allele frequency should have a large confidence interval. Given that a large proportion of cases are compound heterozygous genotypes, this also seems unnecessary to dwell on.

We mainly discuss, in the cited paragraph, variants which have been found in the homozygous state, as they are the most unlikely to be pathogenic.

Half of the 46 TALS, RFMN or LWS families are due to homozygous RNU4ATAC variants (if we consider TALS families only, that’s 20 out of 31).

Minor concerns:

- The text suffers from long sentences with many subordinate clauses. The clarity of the scientific message would benefit from rewriting. This is particularly true of the first and last paragraphs of the introduction, and all descriptions of the cellular assay.

Long sentences were shortened or rephrased in the first and last paragraphs of the introduction and in the result section concerning the cellular assay.

- In all reported cases, can it be confirmed that no other potentially pathogenic variants were identified? More description in the methods as to the process for assessing of the validity of the reported variants is necessary before proceeding to bioinformatic or functional assessment

Whole genome or exome sequencing has been seldom performed, so it is not known whether other potentially pathogenic variants could be identified in some of the reported cases. However, the homogeneity of the clinical signs of the carriers of bi-allelic RNU4ATAC variants and the rarity of these variants in all screened populations make none of the RNU4ATAC publications dubious.

- It is stated that all data are available, but the VCFs for the large cohorts described in the studies should be clearly available for assessment of other potentially pathogenic variants. Are those available from other publications? If so, that information should be provided in the methods

This information is available in the two original publications (Lionel et al, 2018 ; Shaheen et al, 2019).

- It seems that the preferred term is MOPD1 instead of TALS, to prioritize descriptions of pathology rather than named after the describing team

The two terms are widely used; in the two papers published in Science in 2011, one used TALS and the other MOPD I. For our part, we’ve been using TALS in our publications since the beginning.

- A better summary of the unique and overlapping features of the three syndromes could be provided as a supplemental table

Such a summary can be found in the publication by Farach et al (2017). 

- Line 68 references unexplained deaths but there seemed to be neurological etiologies described in the other papers

Death in most cases follows a benign infection, but there is no explanation so far as to why these TALS children die so suddenly. 

- The third paragraph of the introduction needs far more references to support the scientific statements. It would be helpful to specify which tri-snRNPs are disrupted in MOPD1 (line 94-95)

The reference of a review, which provides all the relevant information, was added.

- Line 123 should not have an apostrophe and gnomAD should not be capitalized (here and throughout)

The apostroph has been removed and the two occurrences of GnomAD corrected.

- Line 133 totalising is not a word

We wrote instead “leading to a total number of”

- A specific link to the github mentioned in line 150 should be provided

The github link (https://github.com/cbenoitp/RNU4atac_variants) has been added to the methods.

- Detailed discussion of the domains of U4atac in 222-243 would be better for the introduction than results

We think that the paragraph in question (now lines 271-293) better fits into its present location than in the introduction because the first section of the results consists of an inventory of all RNU4ATAC variants associated with a disease. It is illustrated with a detailed figure. We also discuss in this paragraph variants’ localisation, and summarize the genotypes of patients.

- Similarly, much of the variant description in line 251-269 is highly redundant and could be summarized in a much shorter form

While in the first part of the section, we discuss all RNU4ATAC variants associated with a disease, in the second one, we discuss them according to whether they were found in TALS, RFMN and/or LWS syndromes. We reckon that some parts may appear redundant but we believe that discussing genotypes/phenotypes correlations is expected and important.

- Half as less is not a phrase, in line 275

We changed “half as less” by “half as much”.

- Figure 2A needs to be significantly reformatted to be linear. The zig zag structure makes it as confusing as the text

We have changed the figure according to the recommendations.

- Figure 3A would be more clear as a log-scale x-axis to minimize the white space

We have changed the figure according to the recommendations. 

- Figures 4A/5C are very poor quality and cannot be read

We apologize for the poor quality of the figures displayed in the pdf version of our submitted manuscript. The submitted figures were of much better quality. We further increased the quality and readability of Figures 4A and 5C.

- Figure 5 text is too small to be read

The size of the text in Figure 5 was increased to improve readability.

- In discussion, noncoding genes are not only transcribed long RNAs or short RNAs. Some encode micropeptides, or have other functions. Therefore the word either in line 408 is not appropriate.

We agree that noncoding genes are not restricted to those transcribed in long and short noncoding RNAs, as there is the pseudogene category as well. We have replaced “noncoding genes” by “noncoding RNAs” in the discussion.

- For Figure 6, are there common features of the variants with high vs low correlation? Does that show that the cellular assay has important information about some of the very low CADD scores? If those two low CADD scores are excluded, what is the new R2 value. 

The two variants with low CADD scores, n.42C>G and n.99T>C, are associated with a high splicing efficiency, so there is a very good correlation for them between CADD raw score and splicing efficiency. Those variants for which the correlation between these two parameters is low (n.45A>C, n.65C>T, n.72A>G, n.83G>A and n.124G>A) have all of them a high score and a high splicing efficiency. We discuss in lines 503-508 the likely reason for the high splicing efficiency of n.124G>A. Concerning the other variants, all found in population studies at the heterozygous state, it’s not possible to know if our assay is not sensitive enough to detect their effect or if the CADD predictions in these cases are unreliable. 

Also that figure does not need as many decimal points displayed in the figure.

For the r2 values, the number of decimals displayed on Figure 6 were reduced.

- Figure 1 there is an extra triangle pointing to the U on the right side of the 5-stem loop. What does this signify? 

The triangle on the right side of the 5’ stem loop corresponds to the end of the arrow pointing to the 15.5K protein. We improved the quality of the figure to avoid any confusion.

Also the colors are too similar in Figure 1 – especially the purple and dark blue

We changed the color of the dark purple triangle to a lighter color to make it more distinguishable from the dark blue triangles.

- Abstract the phrase: "… that allows to measure the splicing efficiency of RNU4ATAC variants on a minor (U12-type) reporter intron" needs modification.

We changed the phrase to “… that allows to measure the effect of RNU4ATAC variants on splicing efficiency of a minor (U12-type) reporter intron.”

---

## [Decision Letter · Decision Letter 1]

22 Jun 2020

Clinical interpretation of variants identified in RNU4ATAC, a non-coding spliceosomal gene

PONE-D-20-06355R1

Dear Dr. Sylvie Mazoyer,

We’re pleased to inform you that your manuscript has been judged scientifically suitable for publication and will be formally accepted for publication once it meets all outstanding technical requirements.

Kind regards,

Klaus Brusgaard

Academic Editor

PLOS ONE

Additional Editor Comments (optional):

Reviewers' comments:

Reviewer's Responses to Questions

**Comments to the Author**

1. If the authors have adequately addressed your comments raised in a previous round of review and you feel that this manuscript is now acceptable for publication, you may indicate that here to bypass the “Comments to the Author” section, enter your conflict of interest statement in the “Confidential to Editor” section, and submit your "Accept" recommendation.

Reviewer #2: All comments have been addressed

2. Is the manuscript technically sound, and do the data support the conclusions?

Reviewer #2: Yes

3. Has the statistical analysis been performed appropriately and rigorously? 

Reviewer #2: Yes

4. Have the authors made all data underlying the findings in their manuscript fully available?

Reviewer #2: Yes

5. Is the manuscript presented in an intelligible fashion and written in standard English?

Reviewer #2: Yes

6. Review Comments to the Author

Reviewer #2: The authors have addressed all of my concerns.

7. PLOS authors have the option to publish the peer review history of their article (what does this mean?). If published, this will include your full peer review and any attached files.

Reviewer #2: No

---

## [Editor Report · Acceptance letter]

25 Jun 2020

PONE-D-20-06355R1 

Clinical interpretation of variants identified in RNU4ATAC, a non-coding spliceosomal gene 

Dear Dr. Mazoyer:

I'm pleased to inform you that your manuscript has been deemed suitable for publication in PLOS ONE. Congratulations! Your manuscript is now with our production department. 

Kind regards, 

on behalf of

Dr. Klaus Brusgaard 

Academic Editor

PLOS ONE